# Conversion of the bronchial tree into a conforming electrode to ablate the lung nodule in a porcine model

Izaz Ali Shah [1,10], Hee Yun Seol[2,10], Youngdae Cho [1], Wonjun Ji[3], Jaeyoung Seo[4], Cheolmin Lee[4], Min-Ku Chon[5], Donghoon Shin[6], Justin H. Kim[4], Ki-Seok Choo[7], Junhui Park[8], Juhyung Kim[9], Hyoungsuk Yoo[1,11✉] & June-Hong Kim [5,11✉]

## Abstract

**Background** Radiofrequency ablation (RFA) is one of the treatment options for lung nodules. However, the need for exact delivery of the rigid metal electrode into the center of the target mass often leads to complications or suboptimal results. To overcome these limitations, a concept of conforming electrodes using a flexible material has been tested in this study.

**Methods** A bronchoscopy-guided RFA (CAROL) under a temperature-controlled mode was tested in in-vivo and ex-vivo porcine lungs. Gallium-based liquid metal was used for turning the bronchial tree into temporary RF electrodes. A customized bronchoscopy-guided balloon-tipped guiding catheter (CAROL catheter) was used to make the procedure feasible under fluoroscopy imaging guidance. The computer simulation was also performed to gain further insight into the ablation results. Safety was also assessed including the liquid metal remaining in the body.

**Results** The bronchial electrode injected from the CAROL catheter was able to turn the target site bronchial air pipe into a temporally multi-tined RF electrode. The mean volume of Gallium for each effective CAROL was $0.46 \pm 0.47$ ml. The ablation results showed highly efficacious and consistent results, especially in the peripheral lung. Most bronchial electrodes were also retrieved by either bronchoscopic suction immediately after the procedure or by natural expectoration thereafter. The liquid metal used in these experiments did not have any significant safety issues. Computer simulation also supports these results.

**Conclusion** The CAROL ablation was very effective and safe in porcine lungs showing encouraging potential to overcome the conventional approaches.

## Plain language summary

Lung cancer can be treated by inserting a metal device into the lung via the throat and using this to send radio waves into the cancer. However, using a rigid metal device can cause damage to other areas of the lung and can only treat small cancers. Here, we describe an alternative method to treat lung cancers in which liquid metal is used to fill the spaces within the lung closest to the cancer. We demonstrate that this method can be used to treat cancer in a swine model of lung cancer. Given the positive results we obtained, we think this approach should be tested in a clinical trial in human patients with lung cancer, as it might improve cancer treatment.

[1] Department of Electronic Engineering, Hanyang University, Seoul 04763, Republic of Korea. [2] Department of Internal Medicine, School of Medicine, Pusan National University, Pusan National University Yangsan Hospital, Research Institute for Convergence of Biomedical Science and Technology, Yangsan, Republic of Korea. [3] Department of Pulmonary and Critical Care Medicine, Asan Medical Center, University of Ulsan College of Medicine, Seoul, Republic of Korea. [4] Department of R&D Center, Tau Medical Inc, Busan, Republic of Korea. [5] Department of Cardiology, School of Medicine & Cardiovascular center, Pusan National University & Pusan National University Yangsan Hospital, Yangsan, Republic of Korea. [6] Department of Pathology, School of Medicine, Pusan National University & Pusan National University Yangsan Hospital, Yangsan, Republic of Korea. [7] Department of Radiology, School of Medicine & Medical Research Institute, Pusan National University & Pusan National University Yangsan Hospital, Yangsan, Republic of Korea. [8] Major of Human Bioconvergence, College of Information Technology and Convergence, Pukyong National University, Busan, Republic of Korea. [9] Department of Physiology, University of Toronto, Toronto, Canada. [10] These authors contributed equally: Izaz Ali Shah, Hee Yun Seol. [11] These authors jointly supervised this work: Hyoungsuk Yoo, June-Hong Kim. ✉email: hsyoo@hanyang.ac.kr; junehongk@gmail.com

L ung cancer is the most prevalent oncologic disease[1]. Although open surgery, which comprises lobar resection or pneumonectomy, is the gold standard treatment for lung cancer, these procedures are associated with significant morbidity, and many patients fail to meet the pulmonary physiologic guidelines for open surgery[2]. Stereotactic body radiation therapy (SBRT) is regarded as an alternative modality for patients for whom open surgery cannot be considered. However, the risk of SBRT-associated radiation pneumonitis limits its application[3,4]. Hyperthermic ablation, such as radiofrequency ablation (RFA) or microwave ablation (MWA), is another useful tool for controlling tumors[5]. Compared with SBRT, RFA yields no significant difference in overall survival, especially for tumors smaller than 2 cm[6].

However, the major limitation of the current RFA is its relatively small ablation volume, resulting in a high local recurrence rate for tumors larger than 2–3 cm when a single linear electrode of RFA is applied[7]. Multi-tined electrodes can enlarge the ablation area; however, they reportedly have decreased control and increased invasiveness, which limits their practical use[8,9].

In addition to the effective RFA volume, the way of approach is also related to the overall outcome. To obtain the optimal result, a single RF electrode needs to be positioned at the tumor center in most cases. The CT-guided percutaneous approach makes it highly probable to achieve tumor-central positioning of RF electrodes; but, the complication rate of pneumothorax is up to 50% due to pleural damage[10]. In contrast, the transbronchial approach with bronchoscopy can reduce the pneumothorax rate[11]; however, this benefit is offset by the difficulty in the tumor-central positioning of the rigid electrode (Fig. 1a, b). This limits its applicability only to a subset of tumor patterns, such as the air-bronchus sign[12].

Recently, no-touch ablation was tested in patients with liver cancer, in which at least three fine electrodes were positioned near the periphery of the target tumor (Fig. 1c). This approach showed better local tumor control compared with conventional approach[13–16]. However, the requirement of multiple (at least three) punctures with several electrodes may be too invasive for patients with unfavorable anatomical conditions.

In this study, we aimed to apply the widely used gallium-based liquid metal E-Galn, which has excellent bioavailability, to achieve the same effect as no-touch ablation for lung nodule treatment through an approach. This approach was developed by our group under the project titled, "*Conforming Ablation of Radiofrequency Out of a Liquid Metal (CAROL)*" which involved conducting in vivo and ex vivo ablation tests on porcine lungs, including a pseudotumor model, and performing computational analysis in parallel to better understand the findings (Supplementary Fig. S1). The concept of the 'conforming ablation' has been already proposed in other studies[17–20]. However, the application of the conforming electrode in the air-conducting bronchial tree has not been reported yet. In this study, we would like to propose a practical application of the liquid metal especially in peripheral lung nodule ablative treatment because most lung masses are surrounded by an air-conducting bronchial structure and also the bronchial tree surrounding the peripheral nodule usually is within a certain range of diameter and length that can be turned into an effective temporary RF electrode with liquid metal. In this way, the current limitations of RF ablation due to the use of a rigid electrode can be overcome. Furthermore, 'anatomical ablation of CAROL' along the whole target bronchial tree can have a similar effect to surgical segmentectomy. In our study, we observed that CAROL ablation demonstrates greater efficacy, particularly within pseudotumor models exhibiting characteristic features in gross visual or CT findings. Especially, the effectiveness of CAROL ablation correlates closely with both the bronchial diameter and the length of the targeted area, The liquid metal primarily employed for CAROL ablation could be retrieved without notable safety concerns, By taking specific precautions for the sensitive thoracic regions, extrapulmonary collateral damage can be minimized.

## Methods

**CAROL ablation system and CAROL ablation procedure.** The CAROL system consists of a dedicated CAROL ablation catheter and a bronchial electrode (Fig. 2c, d). The CAROL ablation catheter has a distal balloon occlusion function because the bronchial electrode needs to be confined in a closed space; otherwise, bronchial electrode continuity is easily interrupted due to bronchial structure deformation resulting from tissue edema during ablation. Additionally, its RF electrode is placed inside the lumen to avoid direct contact with bronchial tissue (Supplementary Fig. S2a). The CAROL catheter was also designed to measure the temperature of the bronchial electrode at its distal tip, which reflects the average temperature of the entire ablated tissue. The prototype CAROL catheter was provided by TAU MEDICAL INC. (Busan, Republic of Korea). The catheter has two separate lumens (Supplementary Fig. S2): one is for bronchial electrodes (injection lumen) and the other is for a conventional 0.014" PTCA guidewire (guide wire lumen). The use of the guidewire was up to the discretion of the operator. The main reason for the guidewire use was to facilitate the catheter placement to the target site, especially in the case of tortuous anatomy. And also the other good reason for the guidewire use was to make up the point of discontinuity of the bronchial electrode by tissue edema during the CAROL ablation in some cases, especially with tortuous bronchial anatomy (Supplementary Fig. S3). A commercially available RF generator (M-3004; RF Medical Co., Republic of Korea) was used in this study. Although a variety of ablation modes were tested in each experiment, the temperature-controlled mode (set at 80 °C) was preferably used in our study because of its consistent and effective ablation. CAROL ablation was terminated if any of the following conditions occurred: (1) the impedance rose over 250 Ω or (2) the predetermined time was reached (5, 10, and 15 min according to the experimental plan). If the sudden impedance rise is caused by disruption of bronchial continuity by tissue edema that is evident under the fluoroscopic image (Supplementary Fig. S3), a subtle booster injection of a very small amount of bronchial electrode was very helpful in recovering the continuity of the bronchial electrode. Effective CAROL ablation was defined as a temperature of 60 °C or higher in the temperature of the central bronchial electrode that was measured at the distal tip of the CAROL catheter. Saline and a small amount of contrast dye were infused into the target site of the lung prior to CAROL in the early phase of our study; however, this was not performed afterward because saline filling resulted in irregular ablation at the non-target site. Peripheral lung bronchoscopy (TJF-260V, Olympus, 4 mm outer diameter, 2 mm working channel) was used to guide the CAROL system.

In our study, a gallium-based liquid metal, E-GaIn, consisting of a gallium (75%) and indium (25%) eutectic, was used to create a conformable and atraumatic bronchial electrode. E-GaIn (eutectic of gallium 75% and indium 25%) was provided by Nano Korea Co. (Republic of Korea). As a metal, E-Galn has a high conductivity ($3.4 \times 10^6$ S/m) comparable to that of Pt/Ir electrodes ($4.0 \times 10^6$ S/m), and its thermal conductivity is high enough to make it suitable for use in thermometers. It also has excellent radiopacity, allowing it to be used as a radiocontrast dye. Surprisingly, E-Galn has a very low melting point (15.5 °C), allowing it to maintain its liquid form at room temperature. Because of its excellent radiopacity and high viscosity($2.0 \times 10^{-3}$ Pa s), the bronchial electrode can be fully

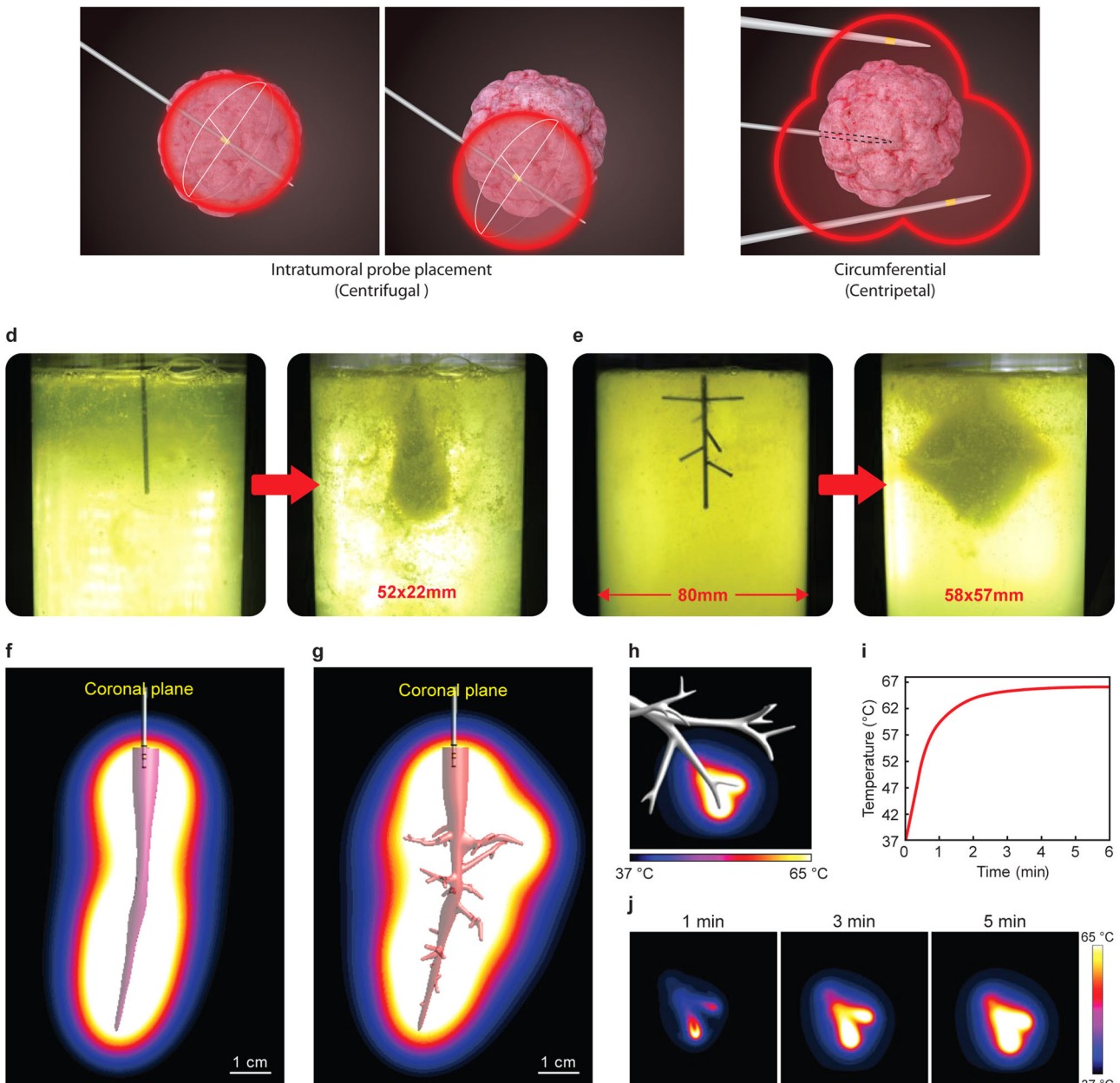

**Fig. 1 Basic concept of CAROL ablation. a, b** Typical illustration of centrifugal ablation that requires tumor center orientation of the ablative electrode.
**c** Typical illustration of centripetal ablation with multiple ablative electrodes. **d, e** Egg white experiment shows that the bronchial structure-shaped
electrode has a larger ablation volume than the straight electrode under the same ablation conditions. **f** Computer simulation of a bronchus filled with
E-GaIn (bronchial electrode) without side branches and **g** with side branches. **h–j** Computer simulation of CAROL ablation showing the ablation spread
pattern according to ablation time. Note that the temperature rise first occurs at sharp tips and then spreads to the proximal parts in CAROL.

controlled under fluoroscopic guidance while being injected into
the target bronchial tree.

**Computer simulation of CAROL.** Finite element analysis was
used to study the efficacy of the use of a liquid metal (E-GaIn) for
RFA. Computer models were built and solved using the finite
element method (FEM) and finite-difference time-domain
(FDTD)-based solvers of commercially available Sim4Life soft-
ware. The classical Penne bioheat transfer equation (Eq. (1)) was
used to determine the temperature distribution in the sur-
rounding tissues and was adopted in this study.

$$\rho c \, \partial T \, \partial t = \nabla \cdot (k \nabla T) + \rho Q + \rho S - \rho b c b \rho \omega (T - Tb) \quad (1)$$

where $T$ is the temperature, $t$ is the time, $\rho$ is the volume density
of the mass, $c$ is the specific heat capacity, $k$ is the thermal con-
ductivity, $Q$ is the metabolic heat generation rate, $\omega$ is the blood
perfusion rate, $S$ is the SAR, $\rho b$ is the density, $cb$ is the specific
heat capacity, and $Tb$ is the temperature of the blood. Sinusoidal
voltages with amplitudes ranging from 50 to 80 V at a frequency
of 480 kHz served as the signals for RFA.

In Eq. (1), $\rho S$, which is denoted as $U$, is used as the heat source
for thermal analysis. The value of $U$ indicates the amount of
power absorbed per unit volume in tissue generated by the
electric field in the EM analysis and is given by

$$U = \frac{\sigma}{2} E^2 \quad (2)$$

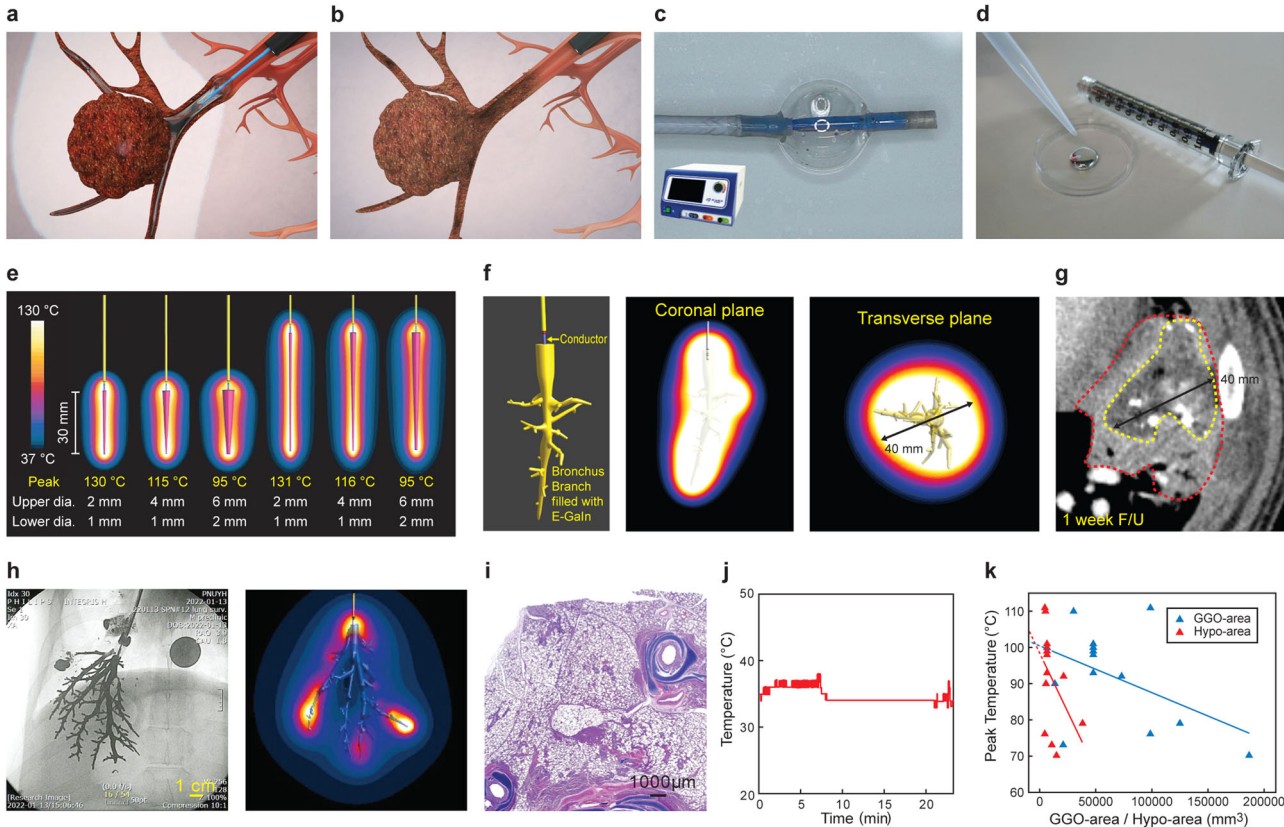

**Fig. 2 CAROL ablation system and the relationship between CAROL ablation and the bronchial diameter and length. a** Denotes the CAROL ablation catheter and the bronchial electrode (E-GaIn) during CAROL ablation. The bronchial electrode is injected through the CAROL catheter to turn the bronchial structure around a mass into the RF electrodes. **b** reveals that the CAROL catheter and the bronchial electrode (E-GaIn) are retrieved from the bronchoscopy(black) after ablation. **c** The CAROL ablation catheter with a balloon at its distal tip & RF generator used in this experiment. **d** E-GaIn was used in this experiment for the bronchial electrode. **e** Computer simulation shows that a bronchus with a smaller diameter, a higher temperature of ablation, and a bronchial length longer than 6 cm leads to an inadequate rise in temperature compared with one that is 3 cm in length. **f** Computer simulation of a real CAROL ablation case shows a similar outcome to the actual data seen on the 1-week follow-up CT image. **g** The hypo-enhanced area (yellow dot circle, **g**) was similar to the area heated by CAROL in the computer simulation (red circle in **f**). The red dot circle of **g** denotes ground glass opacity (GGO). **h** A case of ineffective CAROL in a large territory shows almost normal lung histology (**i**) and no rise in temperature was confirmed through histopathologic examination. H&E staining was used in histopathologic features. **j** during CAROL. The computer simulation also shows a similar finding (**h**). **k** The result shows the relationship between the target area of CAROL & the peak temperature of CAROL in in vivo lung experiments ($r = -0.5$ in GGO and $-0.5$ in hypo-enhancement, $p = 0.12$ and 0.13, respectively; $n = 13$). This graph shows that the larger the target area (expressed by GGO and hypo-enhancement area), the lower is peak temperature.

where $\sigma$ is the electrical conductivity and $E$ is the magnitude of the electric field.

**Animals**. All animals were handled in accordance with the National Institutes of Health guidelines and the Animal Care and Use Committee policies of the Pusan National Yangsan University Hospital (PNUYH), and all animals received humane care. The Institutional Review Board of PNUYH approved all the experimental protocols and studies (IRB no. 2022-020-A1C0). All the animals were euthanized at the end of each experiment.

A total of 22 female pigs (Yorkshire farm swine, weight $46.95 \pm 4.67$ kg) were used for in vivo lung experiments in this study. Ex vivo lung experiments ($n = 25$) were performed in parallel with in vivo experiments (Supplementary Fig. S1). The target sites of the lung and adjacent organs were thoroughly inspected, and the specimens were sent for histopathological examination.

**Ex vivo porcine lung experiments**. Porcine lungs were obtained from a butcher at a nearby market. A piece of the peripheral lung

was used to perform CAROL ablation. In the CAROL ablation procedure, the porcine sirloin muscle was wrapped around a piece of the lung to create a chest wall-like environment. The sirloin muscle, employed to mimic the chest wall, was connected to the ground pad. The CAROL system was inserted into the bronchial lumen of lung segments. The bronchial electrode was then injected through the catheter and its distribution was checked using fluoroscopic imaging (Supplementary Fig. S4). A total of 25 ex vivo lung experiments were performed. For temperature readings of each site, specialized temperature sensing wires (21 models) were placed at each site.

**In vivo porcine lung experiments**. A total of 22 female pigs ($46.95 \pm 4.67$ kg) were used in this study. All the animals were euthanized at the end of the experiment. An Olympus peripheral bronchoscope (TJF-260V, 4 mm in diameter) was inserted into the endotracheal tube and directed to the target sites. All procedures were performed on an X-ray fluoroscopy table (Integris H5000F, Philips Medical Systems) to take advantage of real-time fluoroscopic imaging guidance and cone-beam CT (CBCT) imaging if needed.

**Non-survival experiment**. Non-survival experiments were performed in eight pigs to examine the immediate outcomes of CAROL ablation. Among the eight pigs included, one underwent CBCT immediately after CAROL ablation (Fig. 3).

**Survival experiment**. Fourteen animals underwent CAROL ablation at single or multiple sites (16 sites in total). Follow-up CT imaging was performed at 1, 6, and 12 weeks. At the 12-week follow-up, the animals were euthanized to harvest the lungs for pathological examination. A subgroup of these animals ($n = 8$) underwent regular blood tests to assess the biocompatibility of the CAROL.

In five animals, E-GaIn was injected through the catheter into the bronchus without ablation to examine the pure bronchial electrode effect on the lung and to observe the natural expectoration of the bronchial electrode. In this group, intentionally large amounts of bronchial electrodes were tested in each animal (5 ml in two subjects, 3 ml in two subjects, and 1 ml in one subject), and a deliberately pressurized bronchial electrode

injection that filled the alveolar space was also tested in one animal. All animals in this group were followed up for 1 day, 1 week, and 12 weeks using fluoroscopic imaging, and at the 12-week follow-up, contrast-enhanced CT imaging was also performed. The pigs were euthanized, and their lungs were harvested for pathological examination. All six animals underwent regular blood tests to evaluate the biocompatibility of CAROL.

**Pathologic examination of lung**. After fixation in formalin, the lungs were cut into thin slices to identify ablated lesions or E-GaIn particles. Any lesions or areas containing E-GaIn were then embedded in paraffin blocks. Hematoxylin and eosin staining was performed on these samples according to the standard protocol.

**Creation of the pseudotumor model**. The tumor model was created by mixing the ground porcine muscle with a contrast dye at a 20:1 volume ratio[21,22]. The pseudotumor mix was injected

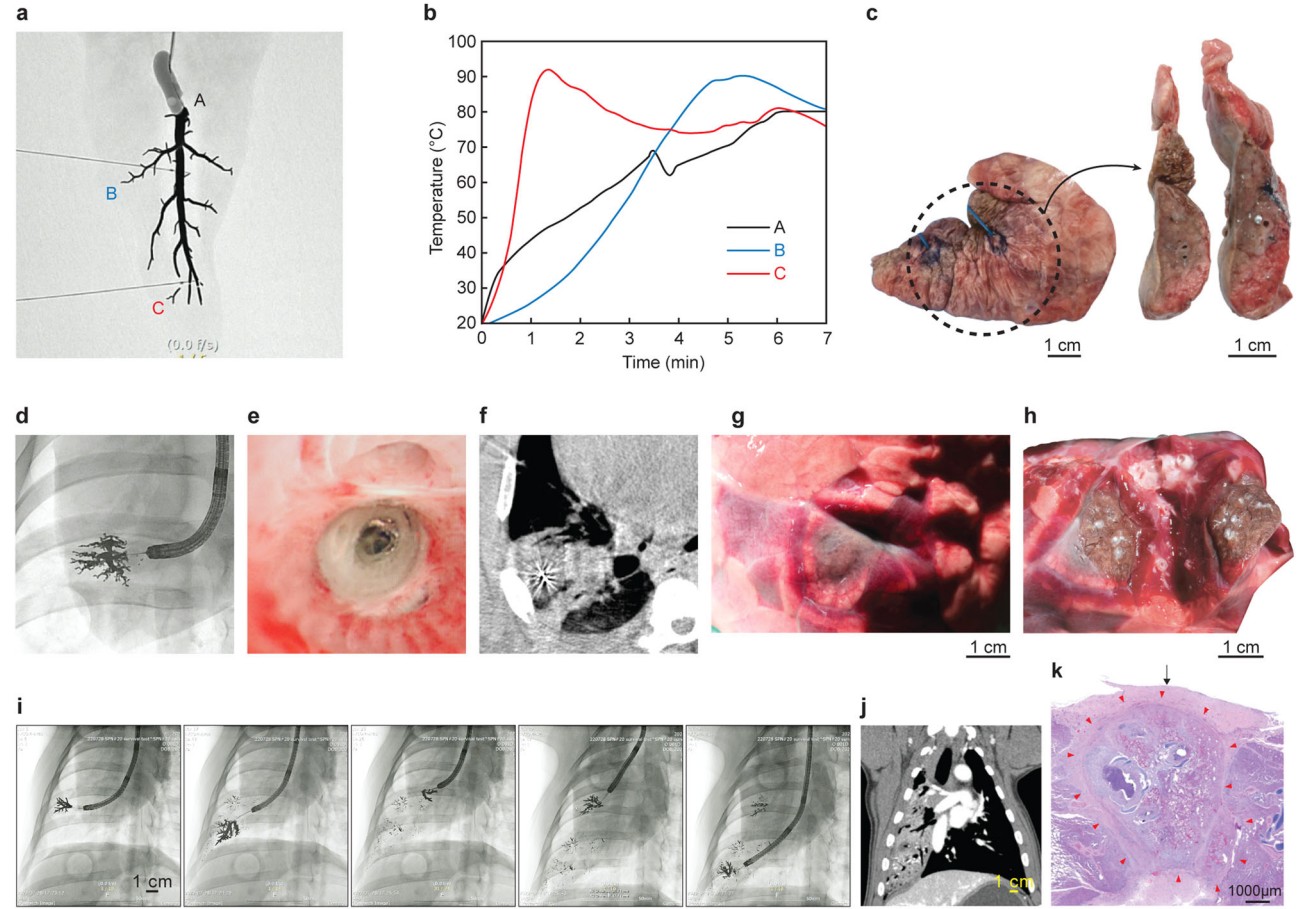

**Fig. 3 CAROL ablation in ex vivo and in vivo porcine lungs.** CAROL ablation in an ex vivo porcine lung segment under a temperature-controlled mode set at 80 °C and gross findings of CAROL ablation. **a** The X-ray fluoroscopic image of a piece of ex vivo porcine lung filled with the bronchial electrode (black tree-like structure) in its bronchus. Multiple temperature sensors (A–C) are placed in the piece of the lung. **b** The rise in temperature begins at the distal area (C) and then spreads to the proximal area. The site where the temperature of the bronchial electrode (central bronchial electrode temperature) is measured is shown by (a). **c** The brown color of coagulation necrosis of the ex vivo lung is evident with a part of the bronchus filled with the bronchial electrode (in the silver-colored dot). **b** In vivo lung with CAROL ablation. **d** Denotes the bronchoscopically guided bronchial electrode injection through the CAROL catheter. **e** Denotes the ablated bronchus seen by bronchoscopy after CAROL ablation. The bronchial electrode in silver is seen in the lumen. **f** Shows the cone-beam CT finding right after CAROL ablation. The lung segment of the corresponding bronchial territory shows ablation injury (anatomical ablation). **g** Shows the target site of CAROL in the harvested lung. **h** Shows the cutting image of the target site. CAROL ablation shows a central brownish necrotic zone with a surrounding hyperemic rim area in the target site. **I, j** Multiple CAROL shots (5 sites) resulted in a large area of ablation seen on CT at the 1-week follow-up. **k** Typical findings of histopathology of CAROL at 12 weeks follow-up. The arrow indicates the thickened and fibrotic visceral pleura. H&E staining was used in histopathologic features.

into the target lung site under fluoroscopic guidance through a Seldinger introducer needle using a percutaneous approach. For CAROL ablation in the pseudotumor model, the CAROL ablation system was placed in the peripheral lung to facilitate the injection of the bronchial electrode into the target site.

**Fluoroscopic imaging guidance, CBCT, and regular follow-up CT imaging**. Fluoroscopic imaging Siemens (Artis zee biplane) was used to obtain fluoroscopic imaging guidance during the CAROL procedures.

*CBCT*: The basic principle of CBCT is the acquisition of multiple X-ray projections during gantry rotation around the volume of interest, usually spanning a total of 200°. The resulting series of images was back-projected to produce a volumetric dataset. Siemens Artis Q (Siemens, Erlangen, Germany) was used with standard CBCT acquisitions provided by the vendor.

*Contrast CT imaging*: All pigs underwent dual-energy CT using a second-generation dual-source CT scanner (Definition Flash; Siemens Healthineers, Forchheim, Germany). The scan ranged from the neck to iliac bifurcation. Non-contrast CT was initially performed using 120 kVp and 100 mA, and then dual-energy CT was acquired using a tube voltage of 100 kVp (tube A) and 140 Sn kVp (tube B). The auto tube current and reference tube current (effective mA) were applied as follows: tube A 150, and tube B, 128. Finally, delay scans were performed at 4 min, 120 kVp, and 160 effective mA. A biphase contrast medium injection protocol was used. First, 115 ml of contrast medium (iomeprol, 300 mg iodine/ml; iomeron 300, Bracco Imaging SpA, Milan, Italy) was injected into the femoral vein (injection rate: 3.5 ml/s), followed by mixed contrast and normal saline (15 + 35 ml). All images were reconstructed using sections.

**Interpretation of ablation injury**. In our study, in addition to direct evidence of histopathologic findings, indirect evidence of coagulation necrosis was assumed to be (1) brown discoloration of the lung tissue on visual estimation[23] or (2) hypo-enhancement zones in the early phase or areas of fibrosis in the late phase of contrast-enhanced lung CT[24].

In this study, the hyperemic areas (red color) surrounding the brownish core area on visual estimation or ground glass opacity (GGO) on contrast-enhanced CT were regarded as the areas that had a mixture of both viable and non-viable tissues.

**E-GaIn specific biocompatibility study**. Blood tests, in addition to CT, were performed not only for CAROL ablation subjects ($n = 3$) but also for subjects who received a bronchial electrode without CAROL ($n = 5$) to investigate the biocompatibility of the bronchial electrode (E-GaIn). Regular CT imaging follow-up was performed at 1 and 12 weeks in all subjects ($n = 8$) with an additional 6-week follow-up CT only in ablated subjects ($n = 3$). Regular serial blood samples were collected preoperatively and 1 day, 1 week, 2 weeks, 4 weeks, 8 weeks, and 12 weeks after the procedures. These blood tests included complete blood count (CBC), liver function test (LFT), renal function test (RFT), electrolyte (Na+, K+), and serum calcium, as well as gallium and indium concentrations. Blood samples were obtained from femoral or subclavian vein puncture after general anesthesia. The analysis was performed at the Clinical Laboratory Center of Pusan National University Yangsan Hospital. Yangsan City, South Korea.

**Measurement of serum gallium and indium concentrations**. Serum gallium and indium concentrations were measured 1 day, 1 week, 2 weeks, 4 weeks, 8 weeks, and 12 weeks after transbronchial E-GaIn injection through the catheter. Blood samples

were sent to a central laboratory (SCL Healthcare, ISO 15189 certified, Seoul, South Korea) and analyzed by inductively coupled plasma-mass spectrometry. Average serum gallium concentration was defined as (the area under the curve of serum gallium concentration from baseline day to peak serum creatinine day)/total number of days from baseline to peak serum creatinine day.

**Measurement of residual bronchial electrodes in the lung by imaging analysis**. The residual bronchial electrode in the lung was measured by the change in the area of the bronchial electrode seen on the X-ray fluoroscopic images.

**Reporting summary**. Further information on research design is available in the Nature Portfolio Reporting Summary linked to this article.

## Results

**Bronchial tree-shaped electrodes with multiple small side branches (CAROL ablation) had a larger ablation volume than the single linear electrode**. Computer simulation, as well as the egg white experiment, showed that a bronchial tree-shaped electrode with multiple side branches resulted in a much larger area of ablation than a single main bronchial electrode without any side branches (Fig. 1d–g). A gallium-based liquid metal (E-GaIn) was used to create the bronchial electrode in the CAROL system (Figs. 2, 3, and supplementary Fig. S2). An increase in temperature was observed in tissues around the distal sharp tip of the electrode first and then spread proximally towards the tissue at the periphery of the electrode (Figs. 1i, 3a, b). Ex vivo lung experiments confirmed the findings of computer simulations (Fig. 3a, b). The temperature of the bronchial electrode measured at the distal tip of the CAROL catheter (central bronchial electrode temperature) reflected the average temperature of the distal and proximal tissues (Fig. 3a, b). A computer simulation of RF ablation with conventional conductive fluid versus CAROL ablation in the same setting had different outcomes in terms of ablation area (Fig. 4g, h). An in-vivo lung experiment that compared saline-associated linear electrode ablation with CAROL in the same subject also showed irregular and less effective coagulation necrosis in saline-associated ablation (Fig. 4a–f).

**CAROL ablation had a typical feature of a central brown zone with a peripheral hyperemic area in the lung**. In vivo CAROL ablation in porcine lungs showed a typical pattern of a central brown zone with a peripheral hyperemic red area (Fig. 3d–h). Most residual bronchial electrodes were found in the central brown zone. In the computer simulation, the area enclosed by the bronchial electrodes exhibited the highest temperature zone (Figs. 1, 2). The typical findings on contrast-enhanced CT were a central hypo-enhancement zone and peripheral ground-glass opacity (GGO)-like lesions. Consistent with computer simulation, the central hypo-enhancement zone was bordered by the bronchial electrode contained in the airway (Fig. 2f). The late CT follow-up (3 months) showed shrinkage of the lesions (Fig. 4a–f), as in other studies[23,24]. Histopathological examination revealed coagulation necrosis in the affected areas surrounded by fibrosed interlobular septa (Figs. 3k, 4f). The alveolar structure was preserved, but the alveolar membranes were covered with fibrin and contained necrotic debris, inflammatory cells, and proteinaceous exudates. E-GaIn particles were mostly found in the bronchi, and less frequently in the parenchyma. Occasionally, there was a fibrotic overlying pleura, but no evidence of significant chronic or ongoing hemorrhage.

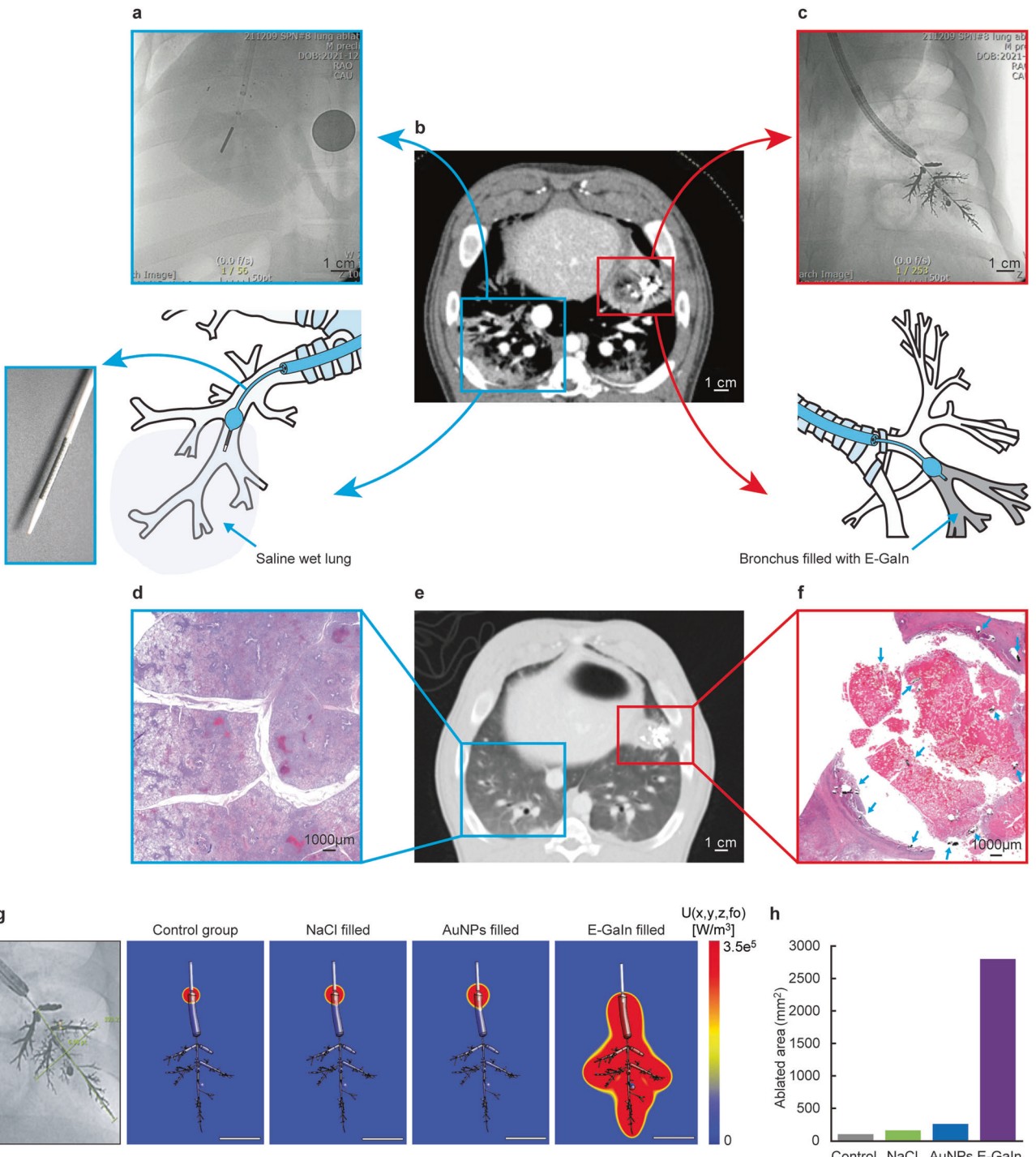

**Fig. 4 Comparison of CAROL ablation with saline-assisted ablation with a linear RF electrode. a** Linear RF coil catheter (TIRA ablation catheter, 15 mm length, internal cooling at a rate of 2 ml/min) ablation for 27 min (temperature control set at 80 °C, total of 9419 W) was performed in the saline-wet lung at the target site. **b** Contrast CT finding at 1 week: the red box denotes the zone of CAROL ablation. The central hypo-enhancement area (dark area) contains the white bright signal of residual E-GaIn in the lung. The hypo-enhancement area is surrounded by peripheral ground glass opacity (GGO). The saline-assisted ablation area (blue box) resulted in irregular ablations. **c** CAROL ablation with 0.5 ml of the bronchial electrode (temperature control set at 80 °C for 7 min, total of 33,636 W). **d** Small scattered irregular foci of ablative injury are seen in the almost normal lung area harvested at 12 weeks follow-up. **e** CT finding at 12 weeks follow-up. **f** Discrete areas of necrosis are surrounded by a well-organized outer fibrotic layer. Scattered residual bronchial electrode particles (black dots indicated by green arrows) are seen within the fibrotic encapsulated area, especially on the margin. H&E staining was used in histopathologic features. **g** Computer simulation results of each RF ablation in the same target area. Conductive fluid containing NaCl or gold nanoparticles (AuNPs) was filled into the target bronchus at the same site of the fluid electrode. **h** Calculated ablation volume from the computer simulation.

**The central bronchial electrode temperature was a surrogate marker for effective CAROL ablation of the target site and the effectiveness of CAROL ablation was closely related to bronchial diameter and length.** Effective CAROL ablation, defined as a bronchial electrode temperature reaching 60 °C or higher, was performed in most cases of CAROL ablations ($n = 25$ out of 28, 89.3%; Supplementary Fig. S1) with a mean hypo-enhancement diameter of $21 \pm 11$ mm and mean GGO-diameter of $32 \pm 18$ mm ($n = 13$ with available 1-week follow-up CT). All pigs that achieved effective CAROL showed at least one of the following coagulation necrosis-associated findings: (1) brown discoloration of the target tissue in the harvested lung immediately after the procedure[23] (Fig. 3h), (2) hypo-enhancement area on contrast CT at the 1-week follow-up assessment (Figs. 2g, 4b), and (3) coagulation necrosis proven by histopathology (Fig. 4f). In the other three pigs that failed to reach 60 °C or higher during CAROL, two of them showed no evidence of coagulation necrosis.

The computer simulation showed that bronchial electrodes placed in bronchi with a slightly enlarged diameter (≥6 mm) or long length (≥6 cm) were less effective in heating up (surrounding tissue (Fig. 2e) as was seen in an in-vivo experiment (Fig. 2h, i). The other important finding was that the volume of GGOs or hypo-enhancement (in 1 week CT data had a nontrivial trend of inverse correlation with the peak temperature of the central bronchial electrode during CAROL ($n = 13$, $r = -0.5$ or $-0.5$, $p = 0.12$ or 0.13, respectively, Fig. 2k). Taken together, if the target area of CAROL is too large, it might not reach the effective ablation temperature (60 °C or higher).

**The extent of the bronchial electrode injection into the target bronchial tree was fully controllable, and most bronchial electrodes were retrievable after the procedure.** The bronchial electrode gradually spread from the proximal part to the distal part without interruption, and its extent of distribution varied depending on the injected volume applied. The operator was able to control the amount and extent of bronchial electrode injection as necessary under fluoroscopic guidance. In the group of effective CAROL ablation in vivo data, the volume of the bronchial electrode was $0.46 \pm 0.47$ ml ($n = 25$). Most of the injected bronchial electrode was retrieved by bronchoscopic suction or natural expectoration over a few days except the small part of the bronchial electrode trapped in a small airway, such as the alveolus (Fig. 5). Fluoroscopic imaging analysis revealed that $87.1 \pm 17.4\%$ of the bronchial electrode was retrieved by active suction or passive expectoration ($n = 14$). In this group, the retrievability of the ablated area (79.0%, $n = 9$) was lower than that of the non-ablated area (95.0%, $n = 5$, $p = 0.02$). The tissue edema of the ablated area and subsequent narrowing of the bronchial lumen may be attributed to this result.

**Ex vivo and in vivo CAROL ablation was effective in porcine lungs with pseudo-tumors.** CAROL ablation for pseudotumor models in the ex vivo lung showed that the tumor tended to receive focused ablation, as indicated by a higher temperature increase than that at any other target site (Fig. 6b–d and Supplementary Fig. S4). The ex vivo data showed that the average temperature of the central bronchial electrode was not different from that of the target lung tissue ($67.9 \pm 7.5$ vs. $64.8 \pm 12.6$ °C, $n = 16$, $p = 0.39$), while the temperature of the pseudotumor was significantly higher than that of the central bronchial electrode ($67.1 \pm 10.7$ vs. $81.9 \pm 15.3$ °C, $n = 5$, $p = 0.01$). In ex vivo ($n = 5$) and in vivo ($n = 3$) pseudo-tumor models, all pseudo-tumors enclosed by the bronchial tree electrode and the pleura showed excellent ablation results (Fig. 6b–f, i, j). However, if a large

portion of the tumor was surrounded by adjacent aerated lung tissue, only the area in contact with the bronchial tree electrode was effectively ablated (Fig. 6g, h).

**Extrapulmonary collateral damage may occur in cases of prolonged ablation or in vulnerable areas.** In the ex vivo lung model, collateral ablative damage was tested in the muscle layer, which wrapped the lung tissue to mimic the chest wall. The depth of damage was proportional to the ablation time ($4.2 \pm 1.5$ mm at 5 min, $12.5 \pm 3.1$ mm at 10 min, $p = 0.01$, supplementary Fig. S5). During CT, chest wall or diaphragmatic ablative damage was observed in all pigs that underwent prolonged CAROL ablation for longer than 10 min ($n = 3$, Fig. 7d–g). Most chest wall injuries did not cause any comorbidities during follow-up, except for one case of late rib fracture (25 min ablation) that did not cause any disabilities, and eventually healed up. Pigs that received CAROL ablation for a short time (<5 min, $n = 14$) did not show any collateral damage.

Among all CAROL ablations in vivo ($n = 28$), two (7.1%) resulted in serious complications. One subject underwent CAROL ablation in the area adjacent to the heart border (mediastinal pleura), which led to myocardial ablative injury and subsequent ventricular fibrillation during prolonged ablation (12 min, Fig. 7h, i). Another subject underwent CAROL ablation solely in the diaphragmatic surface which led to a diaphragmatic hernia after 3 weeks of follow-up, apparently due to ablative injury (Fig. 7j, k).

The computer simulation suggested that, in addition to the duration of the active ablation, the distance from the tip of the bronchial electrode to the pleura, which is fully controllable at the time of E-GaIn injection under fluoroscopic guidance, seemed to be another factor that prevented collateral ablative damage (<5 mm in distance from the tip of the bronchial electrode to the low-temperature zone shown in Fig. 7c and Supplementary Fig. S6). Apart from collateral ablative damage, no other complications, such as pneumothorax, hemothorax, or hemoptysis, were observed, except for one case of infection (3.6%) at the CAROL ablation site.

**The bronchial electrode itself did not lead to any significant safety issues (Fig. 8).** To evaluate the systemic effect of residual bronchial electrodes (E-GaIn) used in CAROL ablation, varying amounts of E-GaIn, up to 5 ml, which is 10 times the usual CAROL dose, were administered by bronchoscopy-guided bronchial transfer. The pigs ($n = 8$) were followed for up to 3 months with serial CT imaging (at 1 week and 12 weeks) and serial blood tests for serum gallium and indium centration as well as assessment of liver and kidney—electrolyte and hematology. The rise of serum gallium and indium concentrations was almost negligible (≤0.03 μg/ml for gallium and ≤0.0016 μg/ml for indium) and peaked within 2 weeks and stabilized after 4 weeks for gallium. Any significant laboratory abnormality suggesting vital organ dysfunction was not noted. Most serum creatinine values ($1.4 \pm 0.4$ in baseline and $1.7 \pm 0.5$ mg/dL in final follow-up, $n = 8$, $p < 0.05$) were within the normal range of porcine serum creatinine (0.75–2.12 mg/dL)[25] except one case of high serum creatinine (2.2 mg/dL) at baseline. The changes of serum creatinine (peak—baseline serum creatinine during follow-up) were not correlated with 'average serum gallium concentration' ($r = 0.4$, $p = 0.3$, Fig. 8). As for the local effect of residual E-GaIn, histopathological examination revealed only a mild inflammatory response with a foreign body reaction around the remaining bronchial electrode in the non-ablated lung.

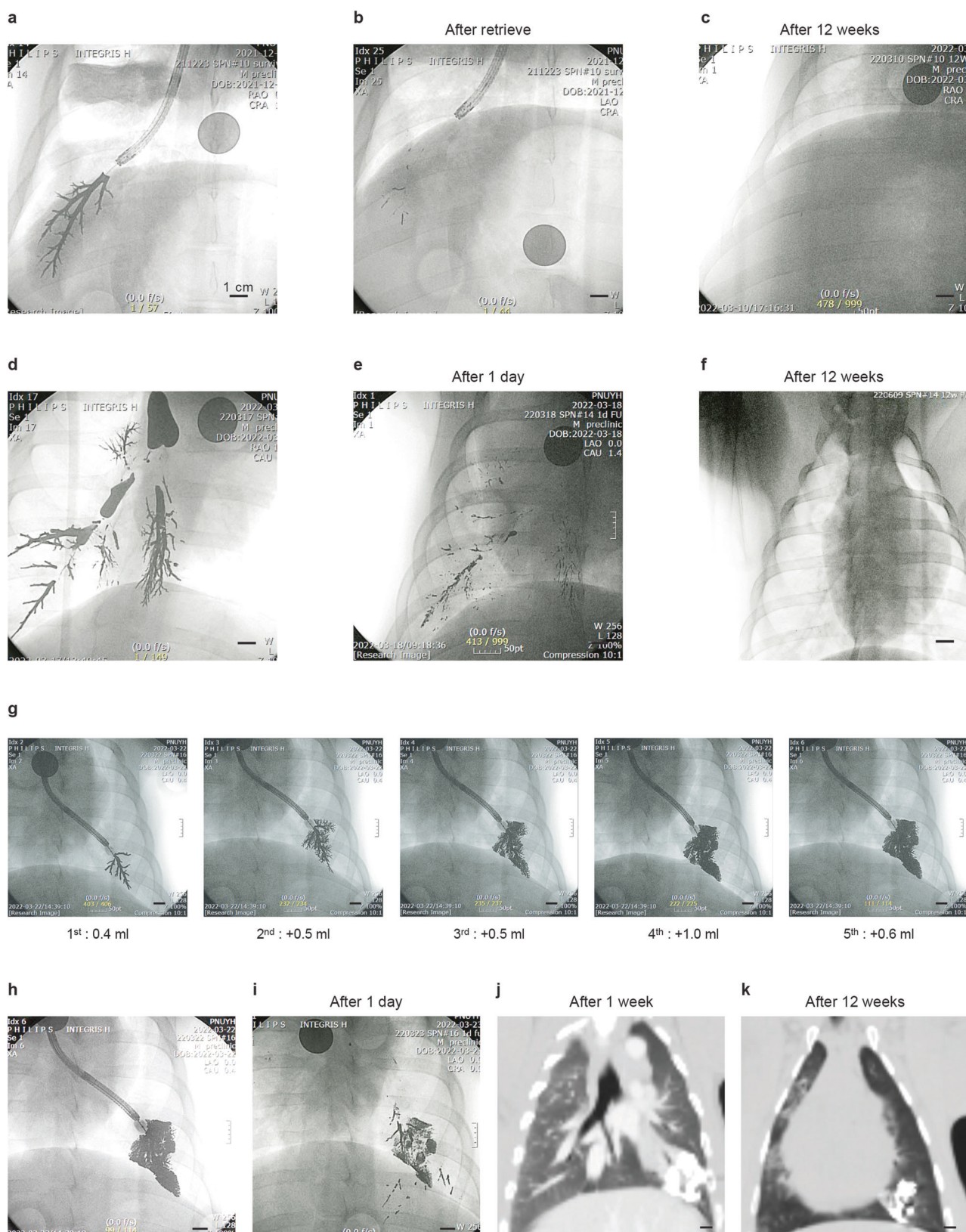

## Discussion

This study showed proof-of-concept evidence and demonstrated the feasibility of "CAROL ablation", an approach for the treatment of pulmonary nodules using medical-grade liquid metal (E-GaIn) as a temporary conforming electrode. The main concept involved replacing the air-pipe bronchial structure with a temporary conforming RF ablation electrode for better efficacy, less invasiveness, and greater simplicity. Thus, the liquidity of the bronchial electrode may offer an important solution to eliminate various problems, such as puncture-related complications, including pneumothorax, related to the invasiveness of percutaneous approaches with rigid electrode needles or multi-tined RF

**Fig. 5 Retrievability of the bronchial electrode (E-GaIn). a** A total of 0.75 ml of the bronchial electrode was injected into the target site for CAROL ablation. **b** Bronchoscopic suction immediately after CAROL ablation. **c** Week 12 follow-up shows further removal of the bronchial electrode by self-expectoration. **d**–**f** A total of 5 ml of the bronchial electrode was injected into the bronchus to examine the efficacy of self-expectoration. Most electrodes were removed by self-expectoration. **g**, **h** The dose (injected volume) dependent expansion of the bronchial electrode ranging from the optimal amount of the bronchial electrode (1st and 2nd shot) up to a forceful high-pressure injection of a total 3 ml of the bronchial electrode performed into the target site. **h**, **i** The alveolar space was filled with bronchial electrodes by this forceful injection, and bronchial electrodes in this small airway zone tended to be trapped. **j**, **k** CT findings show that this excessive residual bronchial electrode material was well confined in its zone without causing any other lung problems.

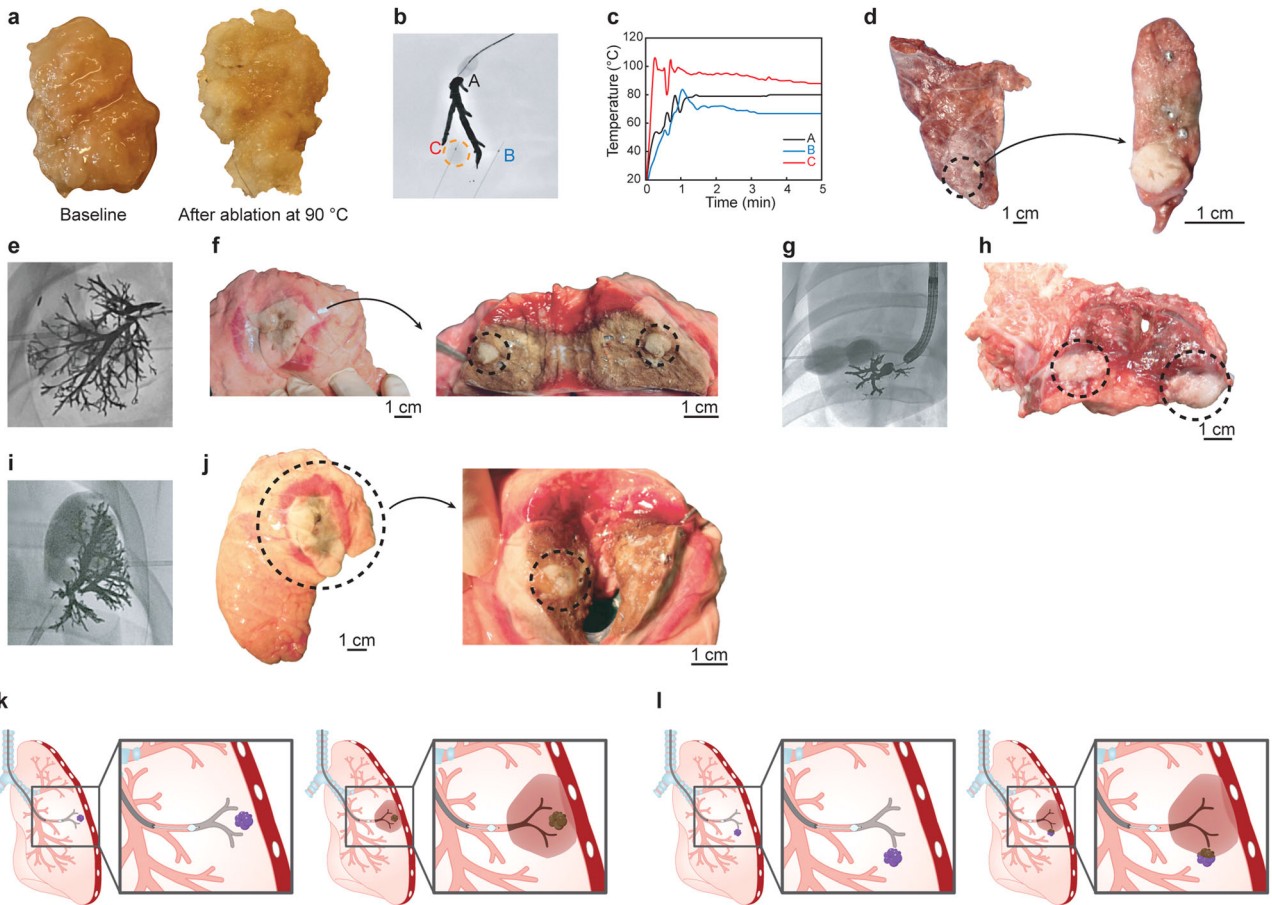

**Fig. 6 Pseudotumor model in CAROL ablation. a** A pseudotumor was made from porcine muscle to be injected through the seldinger needle. **b**, **c** Ex vivo lung experiment with the pseudotumor shows that the tumors had temperatures that were higher than that of the target normal lung tissue during CAROL ablation. **b** The fluoroscopic image of the reverse Y-shaped black tree denotes the bronchial electrode in the piece of ex vivo lung. The round dotted circle is a pseudotumor injected into the piece of ex vivo lung. **c** The temperature of the pseudotumor I is higher than those of B (distal target tissue) and C (proximal target tissue). **d** Gross finding after CAROL ablation. The whitish pseudotumor is well ablated showing similar findings to the right panel of (**a**). **e**, **f**, **i**, **j** CAROL ablation in vivo porcine lung with the pseudotumor injected through the chest wall. **e**, **i** The pseudotumor (round dark one) is surrounded by a black tree with multiple branches (the bronchial electrode). **f**, **j** Harvested lung after CAROL ablation. The whitish pseudotumor is well ablated with surrounding brownish discolored lung tissue. **g**, **h** CAROL ablation of a pseudotumor lying in the margin of the target ablation zone showing incomplete ablation (**h**, dotted circles). **k**, **l** Location of the tumor in relation to the CAROL ablation site. **k** shows the locations of the tumors in panels (**e**, **f**, **j**) in such a situation, CAROL is very effective in tumor control. **l** shows the tumor and the bronchial electrode relation seen in panel (**g**, **h**).

needles with poor controllability, and unwanted damage due to malposition of electrodes.

When a tissue is heated above 42 °C, coagulative necrosis begins to occur. At sufficiently high temperatures (>60 °C), cell death is nearly instantaneous[26]. In our study, "effective CAROL ablation" was defined to have occurred if the central bronchial electrode temperature attained 60 °C or higher based on the excellent thermal conducting characteristic of E-GaIn (9.8 W m$^{-1}$ K$^{-1}$). Lung ablation is dependent on unique energy-tissue interactions. The aerated lung tissue can act as an insulator that limits the conductance of thermal and electric energy. Saline infusion into the target site of the lung could be a solution to this air-insulator challenge. Many studies have examined the enhancing role of ablation volume using conductive fluids such as hypertonic saline (0.113 S/m) or gold nanoparticles (0.138 S/m)[27–29]. However, the E-GaIn used in this study has extremely high conductance (3.4 × 10$^6$ S/m). In addition, E-GaIn showed excellent distribution controllability (bronchial airway only); in contrast, most conductive fluids have poor distribution control, leading to irregular or unpredictable ablation[29,30]. These

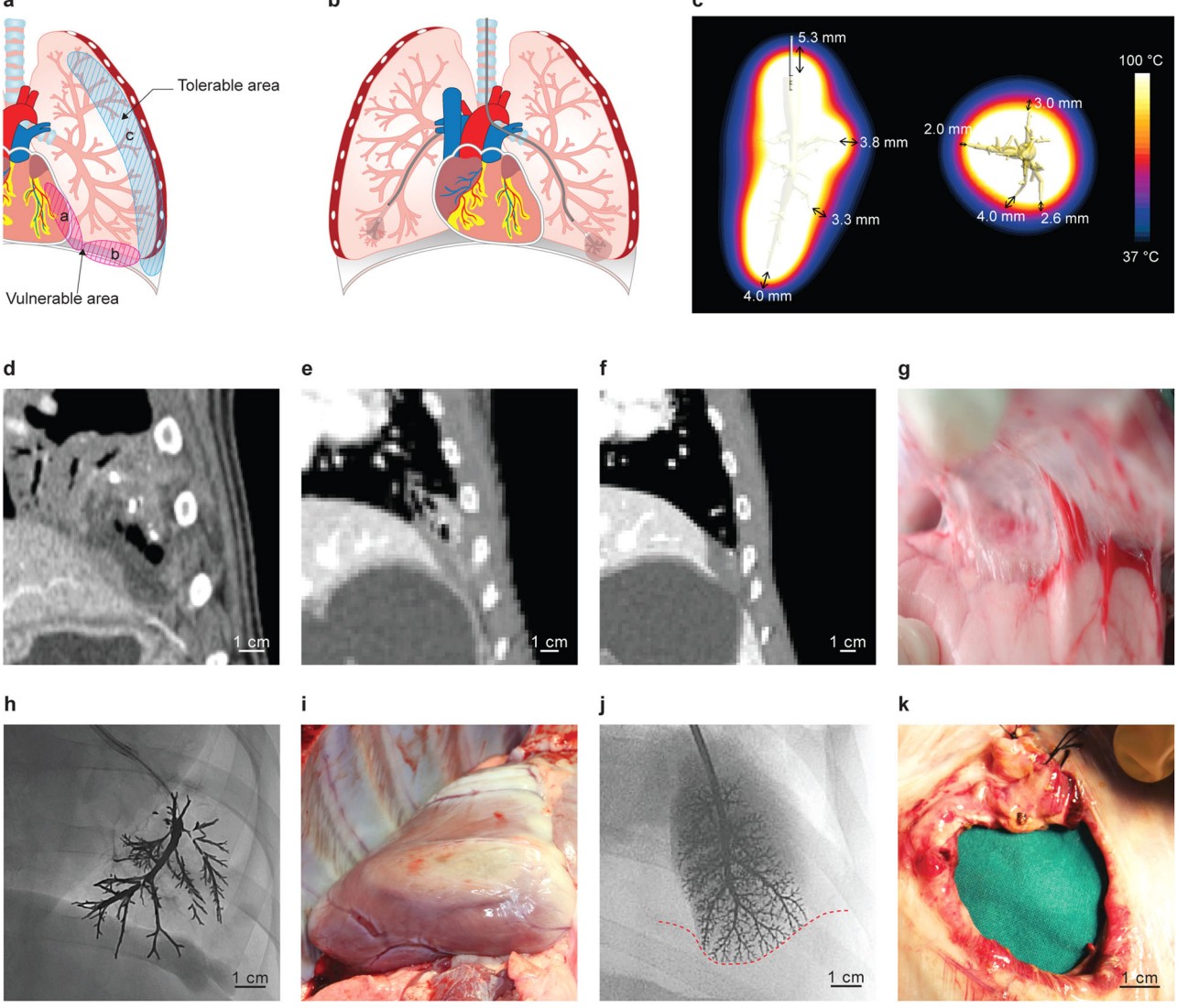

**Fig. 7 Vulnerable area (red oval area) versus tolerable area (blue area) for CAROL. a**, **b** In vulnerable areas, the example of left lung CAROL ablation is more likely to result in collateral damage than in the example of right lung CAROL ablation due to insufficient distancing between the bronchial electrode and the pleura. **c** Computer simulation shows that a distance of at least 5–10 mm from the tip of the bronchial electrode could prevent thermal injury to non-target tissue. **d**–**f** Collateral damage to the pleura and diaphragm in the case of prolonged CAROL ablation (15 min) and its findings at 1 week (**d**), 6 weeks (**e**), and 12 weeks (**f**). **g** Pleural adhesion is observed during dissection at the 12-week follow-up. **h** CAROL ablation reached the left heart boundary which resulted in a burn injury on the surface of the heart (**i**). **j**, **k** In CAROL ablation directed only to the diaphragm surface, without the chest wall, the injected liquid metal reaches the pleural surface (red dotted line), and a diaphragmatic hernia through the defect was observed at the 3-week follow-up, which corresponds to the example of left lung CAROL in (**b**).

differences account for the fact that the role of such conductive fluids is mainly subject to delaying the cessation of RF power due to a sudden increase in electrical impedance[31]. Conversely, the bronchial electrode injected into the bronchial tree for CAROL ablation functioned as an RF electrode.

The main goal of CAROL ablation is to achieve reliable "coagulation necrosis" at the target site. In our study, several indirect indications of coagulative necrosis were found, similar to those in other studies. The first indication was direct visual estimation of the central brownish discoloration of the lung with peripheral hyperemia (red) immediately after ablation (Fig. 3h). In a previous study, lung tissue with brown discoloration after RF ablation was called "necrotic sequestrum"[23]. The peripheral hyperemic zone may contain a mixture of viable and nonviable layers[24]. The central hypo-enhanced area in early-stage CT at 1 week may also represent the area of necrosis (Figs. 2g, 4b), while

the outer GGO-like lesion may have a mixture of viable and nonviable layers[24,32] As shown in our study, ablation according to bronchial territory is a unique feature of CAROL. This approach allows for "anatomical ablation" like surgical segmentectomy (Fig. 3j). Another feature is that CAROL is more likely to be suitable for nodules in peripheral parts of the lungs (the outer one-third of the hilar-costal diameter), where the bronchial diameter is approximately 2–3 mm or less. The lung periphery is the area in which lung nodules most commonly occur[33]. The pulmonary vasculature adjacent to these bronchial systems may be similar in size; therefore, the relatively small vessels (<3 mm) are less likely to interfere with CAROL, as they are known to have a lower heat sinking effect[34–37]. Our study's findings are consistent with those of another study that demonstrated that the diameter of the RF electrode had an inverse relationship with ablation size[38] (Fig. 2e). If large area ablation is required, multisite

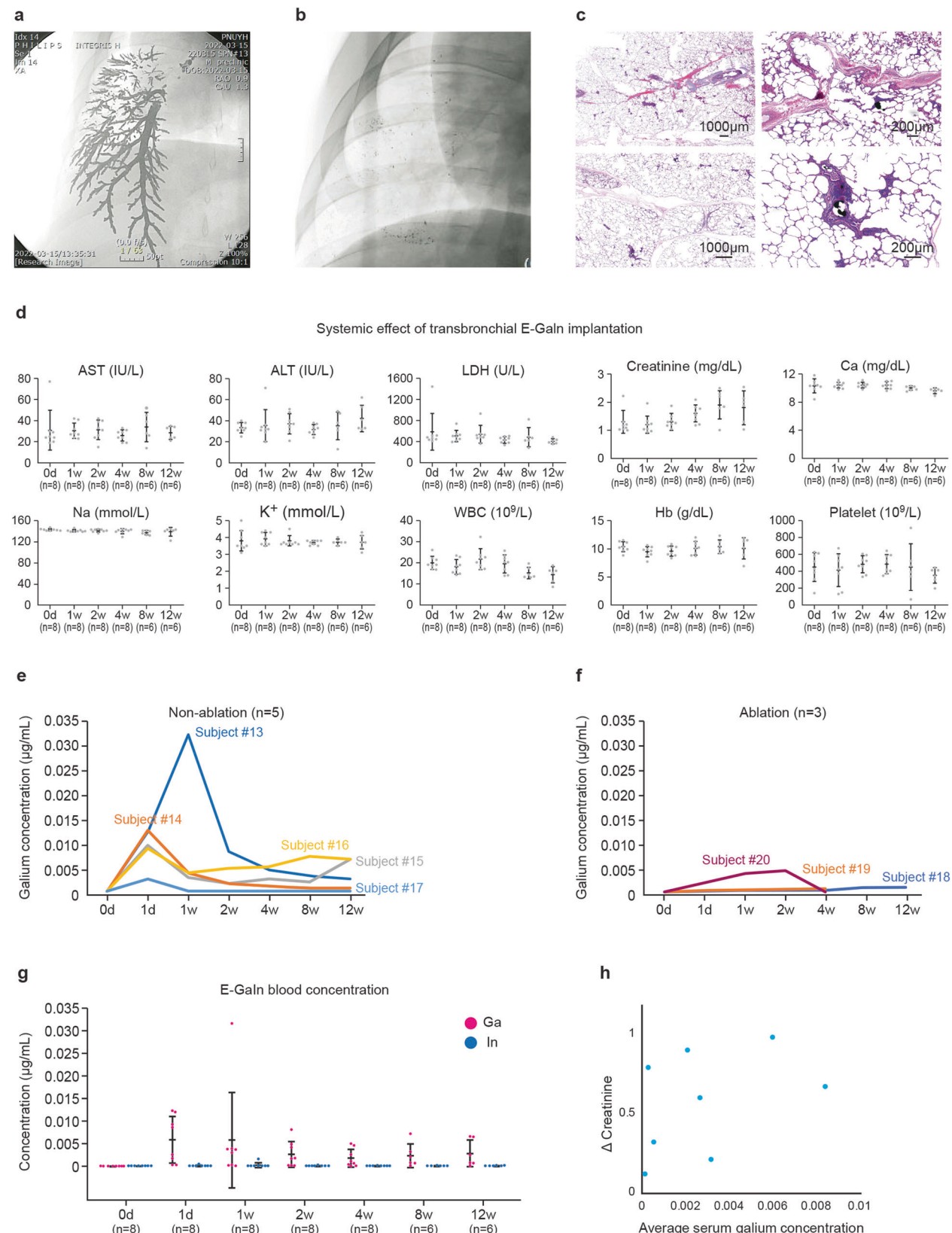

attempts with multiple CAROL ablations should be considered (Fig. 3i, j).

CAROL ablation had a unique centripetal pattern from the periphery to the center (Fig. 1c). This feature has the extraordinary advantage of tumor margin control, as was proven in "no-touch ablation" in liver cancer[13–16]. It is well known that microscopic cancer cell spreading is often seen up to 1 cm outside the tumor margin, which often causes local tumor recurrence in conventional centrifugal RF ablation (Fig. 1a)[39,40]. Therefore, when the bronchial tree surrounding a mass is turned into a temporal electrode, tumor margin control must be excellent. Furthermore, tumor-focused accumulation of ablative energy is a

**Fig. 8 Histological and biochemical analysis. a** An excessive amount of the bronchial electrode (5 ml) was injected intentionally into the lung. **b** Almost no residual bronchial electrode was noted on the fluoroscopic findings at the 12-week follow-up. **c** After harvesting the lung at the 12-week follow-up, histology findings show only mild inflammation in situ along the residual bronchial electrode particle.H&E staining was used in histopathologic features. **d** Conducted an experiment with 8 pigs from baseline to 4 weeks and 6 pigs from baseline to 12 weeks. Serial blood test results showed no significant change in liver function (AST, ALT, LDH), blood components (Hb: hemoglobin, WBC: white blood cells, Platelets), and electrolytes (Na+, K+, Ca: calcium). Most serum creatinine values were within the normal range of porcine serum creatinine (0.75–2.12 mg/dl)[25] except for one case of high serum creatinine (2.2 mg/dl) at baseline. **e**, **f**, **g**, **h** Negligible gallium and indium concentrations were observed in the subjects' blood samples. (5 ml of E-GaIn each in subject #13, 14, 3 ml in subject #15, 3 ml with intentional pressurized injection of E-Gain into alveolar space in subject #16, and 1 ml in subject #17 (*n* = 5), 3.3 ml in subject #18, 0.45 ml in subject #19, 1.42 ml in subject #20 (*n* = 3)), The serum gallium and indium concentrations were almost negligible (≤0.03 μg/ml for Gallium and ≤0.0016 μg/ml for indium) and serum gallium concentration peaked within 2 weeks and stabilized after 4 weeks, regardless of the total amount of bronchial electrodes. **h** The changes in serum creatinine (peak—baseline serum creatinine during follow-up) were not correlated with 'average serum gallium concentration' (*n* = 8, *r* = 0.4, *p* = 0.3).

well-known feature because air-filled lung tissue provides an insulating effect by trapping heat within the target tissue (tumor)[41,42]. In our study, the tumor-focused ablation feature was also preserved in CAROL (Fig. 6b–d and Supplementary Fig. S4).

A high ablation performance may also result in unwanted safety concerns. In our study, the most common CAROL-related injury was intercostal muscle damage at the adjacent chest wall. The intercostal muscle in contact with the CAROL target area appeared to be an outlet for RF energy running through to the ground pad. Although most cases of intercostal muscle ablative damage are self-limiting and tolerable when prolonged (>10 min) CAROL ablation is required, and it should be performed with caution, especially in the vulnerable zone (Fig. 7a, b). Additionally, to avoid unnecessary extrapulmonary injury, as in conventional RF or microwave ablation[5,11], a distance of at least 5 mm from the tip of the bronchial electrode to the pleura or other sensitive organs should be considered (Fig. 7h–k) at the time of the bronchial electrode injection. With these precautions in mind, CAROL ablation can be safely applied to human translation.

Owing to their excellent bioavailability, gallium-based liquid metals have been widely studied in the fields of hyperthermic cancer treatment, artificial organs, and bioelectrodes[42–44]. Gallium itself was used to control cancer or cancer-related problems through parenteral injection[44–49]. For medical applications, E-GaIn can be used in the form of either "bulk material" or "microdroplets". The microdroplet form through the sonication process, in contrast to the bulk form, is related to significant cytotoxicity because a high concentration of gallium and indium ions can be released from the solution[49,50]. Consistently, CAROL ablation, in which bulk-type E-GaIn is used, is associated with almost negligible serum gallium (≤0.03 μg/ml) and indium ≤0.0016 μg/ml) concentrations in pigs, even when excessive amounts of E-GaIn were intentionally applied. This result is also consistent with those of several other studies that investigated the promising effect of direct injection of E-GaIn into tumors for hyperthermic treatment[43,44,50,51]. These researchers performed ISO guideline-directed E-GaIn biocompatibility tests and came up with clear evidence that the injected E-GaIn possessed considerable stability in the body without any leakage of metallic ions into normal organs. According to our study, a single CAROL ablation usually requires less than 1 ml of E-GaIn for the bronchial electrode (0.46 ± 0.47 ml) and a large amount of injected E-GaIn (~70–90%) is going to be retrieved back by active bronchoscopic suction or passive expectoration. Thus, the amount of residual bronchial electrode (E-GaIn) of CAROL is several hundred times less than that remaining after intratumoral injections in the previous studies[43,44,50,51], assuming the same body weight.

Gallium is known to be directly associated with renal toxicity in a dose-dependent manner especially at high serum gallium concentration (10–50 μg/ml)[51,52] but no renal toxicity at low

serum concentration (1 μg/ml)[43,52,53] in humans. The extremely low serum gallium concentration in CAROL should be far from this concern. Among our serial blood tests, slight elevations of serum creatinine within the normal range, which was not related to average serum gallium concentration, appeared to be related to increased muscle mass as the body grows rapidly or with normal age-related change seen in pigs[25,53,54]. Residual bronchial electrodes were not associated with significant lung problems. It is well known that indium is toxic to the lungs[54,55], but this occurs primarily when indium is distributed in the lungs in the form of inhaled gas[54–56]. This is not the case for CAROL. The negligible serum indium concentration of our data is certainly free from in-vivo toxicity[56,57].

In our study, although our data showed a somewhat acceptable result of residual E-Gain in the lung in terms of safety. The E-Gain itself is not a fully biocompatible material and could be a source of toxicity. The long-term consequences of the residual E-Gain in the lung must be investigated with further studies. And also we did not measure the possible change of chemical composition of E-GaIn during the ablation such as oxidation of Gallium[43]. This could have affected the ablation result and or retrievability of the bronchial electrode.

In summary, our study demonstrated the provocative concept of RFA using gallium-based liquid metal for lung nodules with remarkable efficacy and excellent biosafety of the bronchial electrode. Several points were also proposed to keep this technology more secure, such as (1) E-GaIn injection into the optimal size of the bronchial tree and active retrieval of the bronchial electrode after the ablation, (2) ablation time (5–10 min or less) and the concept of distancing of the bronchial electrode to sensitive organ (3) awareness of vulnerable areas versus tolerable areas. Based on these findings, human translation should be strongly encouraged.

**Statistical analysis**. In this study, data are summarized as mean ± SD. In the correlation analysis and its test, Pearson's correlation coefficient was used. When comparing, a paired *t*-test was used for paired data, and otherwise, Welch's *t*-test was used. All statistical analyses were conducted using R (R Statistical Software for Windows, version 4.2.1, 2022, Foundation for Statistical Computing). Statistical significance was set at *p* < 0.05.

## Data availability
The source data for the figures is in Supplementary Data 1. In addition, the experimental data and the CAROL ablation process videos that support the findings of this study are also available in Figshare, accessible at the following identifier https://doi.org/10.6084/m9.figshare.23542374.v1.

## Code availability
Analysis codes required to reproduce the results are available in Figshare, accessible at the following identifier https://doi.org/10.6084/m9.figshare.23542374.v1.

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

## Acknowledgements

This work was supported by the financial grant RS-2023-00232376, 1415187080 from the Korea Institute for Advancement of Technology Evaluation and Planning (KIAT) under the Ministry of Trade, Industry and Energy, Republic of Korea. The funders had no role in the design and conduct of the study, data collection, management, analysis and interpretation, manuscript preparation, review or approval, and the decision to submit the manuscript for publication.

## Author contributions

I.A.S. and Y.D.C. performed all computational analyses of CAROL. H.Y.S. contributed to the design and participated in the in vivo porcine lung experiments and interpretation of data results. Y.J.J. contributed to the analysis of data and manuscript preparation with thoughtful input. J.Y. and C.M. contributed to the in vivo and ex vivo porcine lung experiments, including data gathering of animal data. M.-K.C. contributed to in vivo porcine lung experiments. D.H.S. analyzed all histologic data. J.K. contributed to the design and interpretation of this study in terms of current competing technologies in this field. K.-S.C. contributed to the CT imaging data acquisition and interpretation. J.H.P. analyzed all statistical data. H.S.Y. designed and integrated animal data with computational analysis. J.K. contributed to the re-analysis of data according to the rebuttal work. J.H.K. designed and integrated all data and prepared the manuscript with input from all authors.

## Competing interests

The authors declare the following competing interests: J.-H.K. is a stockholder of TAU Medical Inc. and submitted a patent application related to this project. The remaining authors declare no competing interests.
