## [Peer Review File · Communications Medicine]

Reviewers' comments:

Reviewer #1 (Remarks to the Author):

Thank you very kindly for the opportunity to review this manuscript which describes pre-clinical and modelling studies examining the efficacy of a highly novel RFA delivery system using an injectable electrode

I think this is a very ingenious approach to overcoming the challenges of bronchoscopic delivery of RFA

My main request for the authors is to better describe the RFA device itself. I am not clear on what the bronchial electrode is comprised of – is it multiple metal beads, is it a gelatin solution with suspended metals? How stable is the “electrode” in terms of staying in position/formation during the ablation process? And just as importantly, how is the RFA energy delivered to the “bronchial electrode” – does it require a catheter to be just in direct contact with a proximal component of the “catheter”?

Were there any instances of the bronchial electrode being non-continuous and breaking up within the ablation field so that some component did not receive ablation?

How predictable were the volume of ablation zones? I.e. Given the safety concerns with injury to chest wall/myocardium/diaphragm, can the ablation zone be easily controlled by varying the volume of bronchial electrode material introduced endobronchially?

Title:

Bronchoscopic intervention cannot be considered “non-invasive”. I think the title should be changed to “minimally invasive” or similar wording

Recent clinical studies have demonstrated use of flexible RFA catheters for ablation of pulmonary tumours both with external saline as a conductant (Steinfurt. Respiration 2023) and without (Ishiwata. J Thorac Cardiovasc Surg. 2022). It would be worth citing these papers and speculating on what advantages this technique enjoys over other RFA models in development. These both reported no significant side effects, though achieved smaller ablation zones than the current author achieved. They do however describe a reasonable relationship between energy delivery and ablation zone – is this relevant to the current device?

Reviewer #2 (Remarks to the Author):

The authors proposed an interesting method to use temporary liquid metal for conforming ablation. Liquid metal was delivered via balloon catheter to porcine lungs. The ex vivo results and computer simulation indicated the excellent efficacy with larger ablation volume and focused ablation. The injected soft electrode can also be retrieved by suction and expectoration. The work can be accepted for publication after minor appropriate justification.

The following points are for the authors references:

1 Using liquid and recyclable electrode to expand the performance of rigid electrode is of great significance, while the idea of “conforming ablation” or “conformable treatment” was not the first to be proposed here. There have been researches using liquid metals as soft electrodes/injectable imaging contrast for tumor therapies (Biomaterials, 2017,146,156-167; Minim Invasive Ther Allied Technol, 2018, 27, 233-241; J. Micromech. Microeng. 2018. 28, 034003; IEEE Trans Biomed Eng. 2014, 61, 2161-6). Important citations should be included in the manuscript for a comprehensive introduction.

2 The authors tested the retrievable rate of the liquid metal. From Fig 8a, b, scattered residual of liquid metal can be observed via X-ray images. Could these materials embolize the air pipe?

3 Did the authors measure the retrievability after the ablation or after the injection? The liquid metal electrode may experience oxidization when used as the RFA electrode, thus causing the composition change, such as the increase of gallium oxide. The retrievability would be changed thereafter.

Reviewer #3 (Remarks to the Author):

The authors present an interesting new technology for performing intra-bronchial RFA. While the experiments are neat, the clinical application is a bit of stretch. Most lung nodules for ablation are small, generally located in peripheral lung which would mean they don't abut the bronchial tree as explained in the introduction. Some feedback in no particular order.

1. The technology is interesting in itself and maybe more suitable as an alternate for lobar or segmental resection. Perhaps they should refocus introduction for this and move away from the current goal as an ablative tool for no-touch treatments.

2. The centripetal vs. centrifugal strategies confound basic physics terms. Perhaps they mean circumferential vs. intratumoral probe placement. Suggest rephrasing this throughout the manuscript.

3. The retention of the agent within the lung, even explained away as a small value is considerable and can be a source toxicity. This should be discussed in a nuanced fashion in discussion. The jump to safety for this work is a bit premature.

4. It is unclear how much of the heat transfer effect is from thermal conduction vs. electrical conduction. This can be explained better.

5. Instead of just temperature, the energy deposited per unit volume or thermal dose can be shown in simulation. This is felt to be important as the generator outputs temperature at a fixed range, yet the volume is substantially larger than what can be done when the conductor is in direct contact with the parenchyma.

6. How does things like tissue dessication etc. impact CAROL? If these are problems with RFA, then

how is use of the agent circumventing these challenges?

7. The notation of labeling figures as a1, a2 is extremely confusing. Consider relabeling all figures.

8. How much of CAROL effect has to do with atelectasis?

9. How much of Liquid conductor is getting into the alveolar space? The assumption here is that they are retained in the bronchi. Suspect that might not be the case.

Dear reviewers and editors,

Thank you for the insightful review of our manuscript. The manuscript has been rechecked in accordance with the reviewers' comment. The responses to their comments have been prepared and are given below. And one more author who had contributed to this rebuttal work and the acknowledgement showing a grant support of this study have been added to this revised manuscript.

We thank you and the reviewers for your thoughtful suggestions and insights, which have enriched the manuscript and produced an improved and more balanced account of the research. We hope these revisions and responses are satisfactory, and that the revised manuscript is now suitable for publication in your journal.

Thank you and

Best regards,

June-Hong Kim, MD, PhD

Pusan National University Yangsan Hospital, Yangsan, KOREA

Tel: +82(055)360-1457

Fax: +82(055)360-8784

E-mail: junehongk@gmail.com

Reviewer #1:

My main request for the authors is to better describe the RFA device itself. I am not clear on what the bronchial electrode is comprised of – is it multiple metal beads, is it a gelatin solution with suspended metals?

► I believe that we have provided a specific information of E-GaIn (bronchial electrode) in method. E-GaIn is more likely to be a gelatin solution due to its high viscosity. I would add up the specific information of viscosity to make it clearer. Thank you.

Before

In our study, a gallium-based liquid metal, E-GaIn, consisting of a gallium (75%) and indium (25%) eutectic, was used to create a conformable and atraumatic bronchial electrode. E-GaIn (eutectic of gallium 75% and indium 25%) was provided by Nano Korea Co. (Republic of Korea). As a metal, E-GaIn has a high conductivity (3.4×10^6 S/m) comparable to that of Pt/Ir electrode (4.0×10^6 S/m), and its thermal conductivity is high enough to make it suitable for use in thermometers. It also has excellent radiopacity, allowing it to be used as a radiocontrast dye. Surprisingly, E-GaIn has a very low melting point (15.5 °C), allowing it to maintain its liquid form at room temperature. Because of its excellent radiopacity and high viscosity, the bronchial electrode can be fully controlled under fluoroscopic guidance while being injected into the target bronchial tree

After

In our study, a gallium-based liquid metal, E-GaIn, consisting of a gallium (75%) and indium (25%) eutectic, was used to create a conformable and atraumatic bronchial electrode. E-GaIn (eutectic of gallium 75% and indium 25%) was provided by Nano Korea Co. (Republic of Korea). As a metal, E-GaIn has a high conductivity (3.4×10^6 S/m) comparable to that of Pt/Ir electrode (4.0×10^6 S/m), and its thermal conductivity is high enough to make it suitable for use in thermometers. It also has excellent radiopacity, allowing it to be used as a radiocontrast dye. Surprisingly, E-GaIn has a very low melting point (15.5 °C), allowing it to maintain its liquid form at room temperature. Because of its excellent radiopacity and **high viscosity($2.0 * 10^{-3}$ Pa,s)**, the bronchial electrode can be fully controlled under fluoroscopic guidance while being injected into the target bronchial tree

How stable is the “electrode” in terms of staying in position/formation during the ablation process?

► Thank you for your question. In order to make the ‘stable electrode’ as you pointed out, it is very much important to make the bronchial electrode confined to the target bronchial tree. Thus the occlusion balloon attached to the tip of the CAROL catheter plays the key role to lock up the bronchial electrode. And also, the peripheral bronchial tree only is suitable for CAROL ablation. I think that we already addressed this point in method like following

Method, X paragraph

The CAROL system consists of a dedicated CAROL ablation catheter and a bronchial electrode (Fig. 2c and 2d). The CAROL ablation catheter has a distal balloon occlusion function because the bronchial electrode needs to be confined in a closed space; otherwise, bronchial electrode continuity is easily interrupted due to bronchial structure deformation resulting from tissue edema during ablation.

And just as importantly, how is the RFA energy delivered to the “bronchial electrode” – does it require a catheter to be just in direct contact with a proximal component of the “catheter”?

► Thank you for your question. Yes, continuity of electrical energy transfer from the generator to the bronchial electrode is very much important to perform CAROL ablation. In order to clarify the CAROL system structure, another figure showing how the CAROL system work in the form of flowchart is added in supplementary figure like the followings

Were there any instances of the bronchial electrode being non-continuous and breaking up within the ablation field so that some component did not receive ablation?

► This is very good question. Yes, in some cases, the bronchial electrode were broken up resulting in the interruption of ablation in the distal area from the breaking point of the bronchial electrode. This phenomenon appeared to happen by the tissue edema from ablated injury with immediate rise of impedance. This phenomenon was easily solved by either a booster injection of a very small amount of bronchial electrode or the 0.014 inch guidewire placed in the main bronchial route prior to injection of bronchial electrode as in the following figure.

This point was deemed such a detailed information so we didn't include these descriptions in our original manuscript. But, in response to the reviewer's comment, we have revised it like the following.

Before

Methods, ...

The CAROL catheter was also designed to measure the temperature of the bronchial electrode at its distal tip, which reflects the average temperature of the entire ablated tissue. The prototype CAROL catheter was provided by TAU MEDICAL INC. (Busan, Republic of Korea)

After

The CAROL catheter was also designed to measure the temperature of the bronchial electrode at its distal tip, which reflects the average temperature of the entire ablated tissue. The prototype CAROL catheter was provided by TAU MEDICAL INC. (Busan, Republic of Korea). **The catheter has two**

separate lumens (Fig. S2) : one is for bronchial electrode (injection lumen) and the other is for a conventional 0.014" PTCA guidewire (guidewire lumen). The use of the guidewire was up to the discretion of the operator. The main reason for the guidewire use was to facilitate the catheter placement to the target site especially in the case of tortuous anatomy. And also the other good reason of the guidewire use was to make up the point of breaking up of the bronchial electrode by tissue edema during the CAROL ablation in some cases especially with tortuous bronchial anatomy (Fig. S5).

Before

CAROL ablation was terminated if any of the following condition occurred: (1) the impedance rose over 250 Ω or (2) the predetermined time was reached (5, 10, and 15 min according to the experimental plan)

After

CAROL ablation was terminated if any of the following conditions occurred: (1) the impedance rose over 250 Ω or (2) the predetermined time was reached (5, 10, and 15 min according to the experimental plan). If the sudden impedance rise is caused by disruption of bronchial continuity by tissue edema that is evident under fluoroscopic image (Fig. S5), a subtle booster injection of a very small amount of bronchial electrode was very helpful to recover the continuity of the bronchial electrode.

How predictable were the volume of ablation zones?

► The volume of ablation zone was totally dependent on the bronchial electrode shape and size. The computer simulation showed that the CAROL ablation is effective within a certain range of bronchial electrode length (6 cm or less) and diameter (6 mm or less). With these limitations, CAROL ablation can create an effective ablation volume up to 6 cm diameter in a spheric model depending on 3-dimensional conformation of the bronchial electrode in peripheral lung in which the bronchial diameter is slender. We think that the detailed information about this is fully provided in the Supplementary materials data set with in-vivo and ex-vivo data as well as computational analysis data (Fig. 2e)

Given the safety concerns with injury to chest wall/myocardium/diaphragm, can the ablation zone be easily controlled by varying the volume of bronchial electrode material introduced endobronchially?

► Thank you for the incisive question. Yes, the extent of the bronchial electrode is fully controllable if the operator is injecting the liquid metal slowly under the real time guidance of fluoroscopic imaging. Therefore, when the bronchial electrode reaches out to a point of margin that the operator originally planned, the operator can stop the injection anytime interactively under the fluoroscopic or other image guidance so that the bronchial electrode would not expand beyond the margin. For the clarification of this injection controllability, we revised the manuscript as the following.

Title: Bronchoscopic intervention cannot be considered “non-invasive”. I think the title should be changed to “minimally invasive” or similar wording

▶ Thank you for the incisive question. we will change the title

Before

A novel bronchoscopic guided non-invasive treatment for lung nodule

After

A novel bronchoscopic guided minimally invasive treatment for lung nodule

Recent clinical studies have demonstrated use of flexible RFA catheters for ablation of pulmonary tumours both with external saline as a conductant (Steinfors. *Respiration* 2023) and without (Ishiwata. *J Thorac Cardiovasc Surg.* 2022). It would be worth citing these papers and speculating on what advantages this technique enjoys over other RFA models in development. These both reported no significant side effects, though achieved smaller ablation zones than the current author achieved. They do however describe a reasonable relationship between energy delivery and ablation zone – is this relevant to the current device?

► We would be happy to add up these references with insights of them accordingly

(Add)

12 Steinfors, D. P. *et al.* Safety and feasibility of a novel externally cooled bronchoscopic radiofrequency ablation catheter for ablation of peripheral lung tumours: a first-in-human dose escalation study. *Respiration* **102**, 211-219 (2023).

13 Ishiwata, T. *et al.* Endobronchial ultrasound-guided bipolar radiofrequency ablation for lung cancer: A first-in-human clinical trial. *The Journal of Thoracic and Cardiovascular Surgery* **164**, 1188-1197. e1182 (2022).

(Before)

However, the major limitation of current RFA is its relatively small ablation volume, resulting in a high local recurrence rate for tumors larger than 2–3 cm when a single linear electrode of RFA is applied⁹. Multi-tined electrodes can enlarge the ablation area; however, they reportedly have decreased control and increased invasiveness, which limits their practical use¹⁰⁻¹¹.

(After)

However, the major limitation of current RFA is its relatively small ablation volume, resulting in a high local recurrence rate for tumors larger than 2–3 cm when a single linear electrode of RFA is applied⁹. Multi-tined electrodes can enlarge the ablation area; however, they reportedly have decreased control and increased invasiveness, which limits their practical use¹⁰⁻¹³.

► They do however describe a reasonable relationship between energy delivery and ablation zone – is this relevant to the current device?

Based on the analysis of our experimental data ($n=8$, $r=0.28$, $p=0.53$), it can be concluded that the correlation between the ablation volume and energy is not statistically significant. The limited sample size in our experiment may account for this poor correlation between these variables. Therefore, further research with a larger sample size is warranted to obtain a more conclusive understanding of the relationship between ablation volume and energy.

Reviewer #2:

The authors proposed an interesting method to use temporary liquid metal for conforming ablation. Liquid metal was delivered via balloon catheter to porcine lungs. The ex vivo results and computer simulation indicated the excellent efficacy with larger ablation volume and focused ablation. The injected soft electrode can also be retrieved by suction and expectoration. The work can be accepted for publication after minor appropriate justification.

The following points are for the authors references:

Using liquid and recyclable electrodes to expand the performance of rigid electrodes is of great significance, while the idea of “conforming ablation” or “conformable treatment” was not the first to be proposed here. There have been researches using liquid metals as soft electrodes/injectable imaging contrast for tumor therapies (Biomaterials, 2017,146,156-167; Minim Invasive Ther Allied Technol, 2018, 27, 233-241; J. Micromech. Microeng. 2018. 28, 034003; IEEE Trans Biomed Eng. 2014, 61, 2161-6). Important citations should be included in the manuscript for a comprehensive introduction.

► Thank you for the comments. According to this valuable comment, we have changed the description of CAROL introduction like the following.

In this study, we aimed to apply the widely used gallium-based liquid metal E-Galn, which has excellent bioavailability, to achieve the same effect as no-touch ablation through a novel approach. This novel approach was developed by our group under the project titled, “Conforming Ablation of Radiofrequency Out of a Liquid metal (CAROL)” that involved conducting in vivo and ex vivo ablation tests on porcine lungs, including a pseudotumor model, and performing computational analysis in parallel to better understand the findings. To the best of our knowledge, this is the first approach that uses liquid metal as a flexible temporary electrode for RFA. As most lung masses are surrounded by an air-conducting bronchial structure, the bronchial tree surrounding the mass can be turned into a temporary RF electrode using a liquid metal. In this way, the current limitations of RF ablation due to the use of a rigid electrode can be overcome.

In this study, we aimed to apply the widely used gallium-based liquid metal E-Galn, which has excellent bioavailability, to achieve the same effect as no-touch ablation **for lung nodule treatment**

under the project titled, “Conforming Ablation of Radiofrequency Out of a Liquid metal (CAROL)” that involved conducting in vivo and ex vivo ablation tests on porcine lungs, including a pseudotumor model, and performing computational analysis in parallel to better understand the findings. The concept of the ‘conforming ablation’ has been already proposed in other studies(Biomaterials, 2017,146,156-167; Minim Invasive Ther Allied Technol, 2018, 27, 233-241; J. Micromech. Microeng. 2018. 28, 034003; IEEE Trans Biomed Eng. 2014, 61, 2161-6). However, the application of the conforming electrode in the air-conducting bronchial tree has not been reported yet. In this study, we would like to propose a practical application of the liquid metal especially in peripheral lung nodule ablative treatment because most lung masses are surrounded by an air-conducting bronchial structure and also the bronchial tree surrounding the peripheral nodule usually is within a certain range of diameter and length that can be turned into a effective temporary RF electrode with liquid metal. In this way, the current limitations of RF ablation due to the use of a rigid electrode can be overcome.

The authors tested the retrievable rate of the liquid metal. From Fig 8a, b, scattered residual of liquid metal can be observed via X-ray images. Could these materials embolize the air pipe?

▶ Thank you for the question. Yes, most residual liquid metal was entrapped in the small lumen of bronchus or alveoli in the target area. Because the target area is intended to destroy, air-pipe embolization in situ would not have any clinical problem. We didn't find any case of heterotrophic embolization of other big air bronchial pipe even with excessive amounts of liquid metal injection in our study. we believe that this point is already addressed in the result.

Did the authors measure the retrievability after the ablation or after the injection? The liquid metal electrode may experience oxidization when used as the RFA electrode, thus causing the composition change, such as the increase of gallium oxide. The retrievability would be changed thereafter

► Thank you for the really valuable comment. In fact, we were not aware of this point/ In our study, E-gaIn was taken care of in a very similar way to common medical device handling. The E-GaIn was kept sealed off until it is used for the experiment. So we believe that the chance of oxidization may be very low. But unfortunately, we did not check the point due to lack of our lab's facility. Anyway, the retrievability data after ablation or after simple injection was like the followings.

As the reviewer pointed out, the retrievability is lower in the group of ablation. This result may be explained by the composition change of E-GaIn as the reviewer commented or by more entrapment of the liquid metal from the local tissue edema in the ablation area. We have made revision on our manuscript to make this point clear

	Ablation (n=9)	Injection only (n=5)
Retrievability (%)	79.0%	95.0%

Before

Result

....Most of the injected bronchial electrode was retrieved by bronchoscopic suction or natural expectoration over a few days except the small part of bronchial electrode trapped in a small airway, such as the alveolus (Fig. 5). Fluoroscopic imaging analysis revealed that $82.3 \pm 23.2\%$ of the bronchial electrode was retrieved by active suction or passive expectoration (n=21).

After

Most of the injected bronchial electrode was retrieved by bronchoscopic suction or natural expectoration over a few days except the small part of bronchial electrode trapped in a small airway, such as the alveolus (Fig. 5). Fluoroscopic imaging analysis revealed that $87.1 \pm 17.4\%$ of the bronchial electrode was retrieved by active suction or passive expectoration (n=14). In this group, the retrievability of the ablated area (79.0%, n=9) was lower than that of the non-ablated area (95.0%, n=5, p=0.02). The tissue edema of the ablated area and subsequent narrowing of bronchial lumen may be attributed to this result.

Discussion

Study limitations

In our study, we did not measure the possible change of chemical composition of E-GaIn during the ablation such as oxidation of Gallium (Wang, D. *et al.* Non-Magnetic Injectable Implant for Magnetic Field-Driven Thermochemotherapy and Dual Stimuli-Responsive Drug Delivery: Transformable Liquid Metal Hybrid Platform for Cancer Theranostics. *Small* **15**, e1900511 (2019). This could have affected the ablation result and/or retrievability of the bronchial electrode.

The authors present an interesting new technology for performing intra-bronchial RFA. While the experiments are neat, the clinical application is a bit of stretch. Most lung nodules for ablation are small, generally located in peripheral lung which would mean they don't abut the bronchial tree as explained in the introduction. Some feedback in no particular order.

► Thank you for the comment. Most peripheral lung nodules are known to abut the bronchial tree like the following figure. And also, for making an effective CAROL ablation, the bronchial tree does not necessarily contact the mass directly as was shown in our ex-vivo experiment

(from “Vol 7, Supplement 4 (December 20, 2015): Journal of Thoracic Disease (The Past, Current and Future Development of High Technology Applied in Thoracic Diseases))

The technology is interesting in itself and maybe more suitable as an alternate for lobar or segmental resection. Perhaps they should refocus introduction for this and move away from the current goal as an ablative tool for no-touch treatments.

► Thank you for the comment. We would like to revise the introduction accordingly like the followings

Before

. As most lung masses are surrounded by an air-conducting bronchial structure, the bronchial tree surrounding the mass can be turned into a temporary RF electrode using a liquid metal. In this way, the current limitations of RF ablation due to the use of a rigid electrode can be overcome.

After

. As most lung masses are surrounded by an air-conducting bronchial structure, the bronchial tree surrounding the mass can be turned into a temporary RF electrode using a liquid metal. In this way, the current limitations of RF ablation due to the use of a rigid electrode can be overcome.

Furthermore, 'anatomical ablation of CAROL' along the whole target bronchial tree can have a similar effect to surgical segmentectomy.

The centripetal vs. centrifugal strategies confound basic physics terms. Perhaps they mean circumferential vs. intratumoral probe placement. Suggest rephrasing this throughout the manuscript.

► Thank you for the comment. However, the concept of centripetal ablation is already used in no-touch ablation. as followings. Anyway, we would like to comply with the reviewer’s comment by adding the ‘circumferential’ and ‘intratumoral’ to the ‘centrifugal’ and ‘centripetal’ respectively.

Ref. 1 No touch radiofrequency ablation for hepatocellular carcinoma: a conceptual approach rather than an iron law. Hepatobiliary Surg Nutr. 2022 Feb; 11(1): 132–135

In comparison with multi-monopolar, multi-bipolar mode allows much more energy deposition inside than outside the arrangement of electrodes and therefore to perform the ablation from the periphery to the center of this arrangement (**centripetal ablation**) and not to every direction from each applicator (**centrifugal ablation**) (Figure 1) (12).

Ref. 2 Outcome of No-Touch Radiofrequency Ablation for Small Hepatocellular Carcinoma: A Multicenter Clinical Trial. <https://doi.org/10.1148/radiol.2021210309>

During the no-touch RFA technique, multiple electrodes are inserted outside of the tumor boundary and sequentially activated. Therefore, thermal energy is first deposited outside the tumor margin, and energy is thereafter delivered **centripetally** to the target tumor

The retention of the agent within the lung, even explained away as a small value is considerable and can be a source toxicity. This should be discussed in a nuanced fashion in discussion. The jump to safety for this work is a bit premature

► Thank you for the comment. We respectfully accept the reviewer’s comment. So the paragraph of ‘study limitations’ has been added up in the last part of discussion

Study limitations

Although our data showed a somewhat acceptable result of residual E-Gain in the lung in terms of safety. The E-Gain itself is not fully biocompatible material and could be a source of toxicity. The long term consequences from the residual E-Gain in the lung must be investigated with further studies

It is unclear how much of the heat transfer effect is from thermal conduction vs. electrical conduction. This can be explained better.

► In equation (1), ρS , which is denoted as U is used as the heat source for thermal analysis. The value of U indicates the amount of power absorbed per unit volume in tissue generated by the electric field in the EM analysis and is given by

$$U = \frac{\sigma}{2} E^2 \quad (2)$$

where σ is the electrical conductivity and E is the magnitude of the electric field.

Before

Finite element analysis was used to study the efficacy of the use of a liquid metal (E-Galn) for RFA. Computer models were built and solved using the finite element method (FEM) and finite-difference time-domain (FDTD)-based solvers of commercially available Sim4Life software. The classical Pennes bioheat transfer equation (Equation 1) was used to determine the temperature distribution in the surrounding tissues and was adopted in this study.

$$\rho c \frac{\partial T}{\partial t} = \nabla \cdot (k \nabla T) + Q + \omega \rho_b c_b (T_b - T) \quad (1),$$

where T is the temperature, t is the time, ρ is the volume density of the mass, c is the specific heat capacity, k is the thermal conductivity, Q is the metabolic heat generation rate, ω is the blood perfusion rate, S is the SAR, ρ_b is the density, c_b is the specific heat capacity, and T_b is the temperature of the blood. Sinusoidal voltages with amplitudes ranging from 50 to 80 V at a frequency of 480 kHz served as the signals for RFA.

After

Finite element analysis was used to study the efficacy of the use of a liquid metal (E-Galn) for RFA. Computer models were built and solved using the finite element method (FEM) and finite-difference time-domain (FDTD)-based solvers of commercially available Sim4Life software. The classical

Penne bioheat transfer equation (Equation 1) was used to determine the temperature distribution in the surrounding tissues and was adopted in this study.

$$\rho \frac{\partial T}{\partial t} = \rho \cdot (\nabla^2 T) + Q + \omega \rho_b (T_b - T) - \rho S \quad (1),$$

where T is the temperature, t is the time, ρ is the volume density of the mass, c is the specific heat capacity, k is the thermal conductivity, Q is the metabolic heat generation rate, ω is the blood perfusion rate, S is the SAR, ρ_b is the density, c_b is the specific heat capacity, and T_b is the temperature of the blood. Sinusoidal voltages with amplitudes ranging from 50 to 80 V at a frequency of 480 kHz served as the signals for RFA.

➤ In equation (1), ρS , which is denoted as U is used as the heat source for thermal analysis. The value of U indicates the amount of power absorbed per unit volume in tissue generated by the electric field in the EM analysis and is given by

$$U = \frac{\sigma}{2} E^2 \quad (2)$$

where σ is the electrical conductivity and E is the magnitude of the electric field.

Instead of just temperature, the energy deposited per unit volume or thermal dose can be shown in simulation. This is felt to be important as the generator outputs temperature at a fixed range, yet the volume is substantially larger than what can be done when the conductor is in direct contact with the parenchyma.

► Generally, both thermal and electrical conduction contribute to the heat transfer effects in RFA, and the final ablation zone is created by a combination of what has been termed direct heating (equation 2), and thermal diffusion of that heat into lower temperature tissues at the ablation periphery. The amount of heat generated by the direct heating is usually small and in close proximity to the applicator (18 mm radial)¹, however, the specific contribution of thermal and electrical conduction to the heat transfer effects in RFA strongly depends on a variety of factors, such as the frequency and power of the radiofrequency energy used, the electrical properties of the tissue, and the specific RFA technique employed, and more importantly the shape of the electrode.

Although the used simulation software Sim4life has limitations to calculate the individual heat transfer effect due to direct heating and thermal conduction, comparing the power deposition against temperature distribution in Fig. S_y (supplementary figure) shows that most of the heat transfer is owing to direct heating (U). Direct heating can be controlled by adjusting the applicator design and power input, potentially creating larger ablations in shorter times. From a clinical perspective, greater direct heating may improve the predictability and efficacy of RF ablation. Our findings suggest that utilizing EGaIn-filled bronchus branches leads to a larger volume of direct heating, indicating that electrodes with broader heating patterns may be preferable, particularly for larger lesions.

1. Schramm, W., Yang, D., Wood, B. J., Rattay, F. & Haemmerich, D. Contribution of direct heating, thermal conduction, and perfusion during radiofrequency and microwave ablation. *The Open Biomedical Engineering Journal* 1, 47–52 (2007).

How does things like tissue dessication etc. impact CAROL? If these are problems with RFA, then how is use of the agent circumventing these challenges?

▶ You for the question which is a very important point of RF ablative therapy. In our study, we intentionally used the temperature controlled ablation instead of ‘energy controlled ablation’ especially to mitigate the roll off phenomenon of RF ablation. Under the temperature controlled mode, we didn’t observe any abrupt premature roll-off phenomenon in our experiments. And it was described in the methods already like the following . Thank you.

.....Although a variety of ablation modes were tested in each experiment, the temperature-controlled mode (set at 80 °C) was preferably used in our study because of its consistent and effective ablation.
.....

The notation of labeling figures as a1, a2 is extremely confusing. Consider relabeling all figures.

▶ Thank you for the comments, we have changed the labelings accordingly.

How much of CAROL effect has to do with atelectasis?

▶ Thank you for the questions. As was shown in the findings of (1) bronchoscopic image of the ablated bronchus (Fig. 3) and (2) the CT finding that showed obliteration of bronchial lumen (Fig. 4) in the target site, the effective CAROL ablation was more likely to present with the obliteration of bronchus in the target site that must lead to ‘atelectasis’. We believe that this finding appears good to lower the risk of bronchopleural fistula that is a rare but fatal complication of the current available lung ablation therapy.

How much of Liquid conductor is getting into the alveolar space? The assumption here is that they are retained in the bronchi. Suspect that might not be the case

▶ Thank you for the questions. As was shown in the findings of (1) bronchoscopic image of the ablated bronchus (Fig. 3) and (2) the CT finding that showed obliteration of bronchial lumen (Fig. 4) in the target site,

As you can see, we performed extensive additional analysis and tried our best to address each of the issues raised by the reviewers. We hope that these revisions have strengthened our manuscript so that it better meets the requirements of your prestigious journal.

Sincerely yours,

Dr. June-Hong Kim, MD., PhD.

Division of Cardiology, Department of Internal Medicine, Pusan National University Yangsan Hospital,
Medical School of Pusan National University, Yangsan, South Korea.

E-mail: junchongk@gmail.com.

End of authors' reply

REVIEWERS' COMMENTS:

Reviewer #2 (Remarks to the Author):

The authors well addressed the review comments and suggestions. The revised article is in a nice format. I would like to recommend accept the work publication.

Reviewer #3 (Remarks to the Author):

Good work on revising the manuscript.